

# Assessment of potential seismic hazard for sensitive facilities by applying seismo-tectonic criteria: an example from the Levant region

Matty Sharon[1,2], Amir Sagy[1], Ittai Kurzon[1], Shmuel Marco[2], Marcelo Rosensaft[1]

1. Geological Survey of Israel, Jerusalem 9371234, Israel

2. Porter School of the Environment and Earth Sciences, Tel Aviv University, Tel Aviv 6997801, Israel

*Correspondence to*: Amir Sagy (asagy@gsi.gov.il)

## Abstract

We present a methodology for mapping faults that constitute a potential hazard to structures, with an emphasis on special facilities such as dams and nuclear power plants. The methodology categorises faults by hierarchical seismo-tectonic criteria, which are designed according to the degree of certainty for recent activity and the accessibility of the information within a given region. First, the instrumental seismicity is statistically processed to obtain the gridded seismicity of the earthquake density and the seismic moment density parameters. Their spatial distribution reveals the zones of the seismic sources, within the examined period. We combine these results with geodetic slip rates, historical earthquake data, geological maps and other sources to define and categorise faults that are likely to generate significant earthquakes (M ≥ 6.0). Their mapping is fundamental for seismo-tectonic modelling and for PSHA analyses. In addition, for surface rupture hazard, we create a database and a map of capable faults, by developing criteria according to the regional stratigraphy and the seismotectonic configuration. The relationship between seismicity slip dynamics and fault activity through time is an intrinsic result of our analysis that allows revealing the tectonic evolution of a given region. The presented methodology expands the ability to differentiate between subgroups for planning or maintenance of different constructions or for research aims, and can be applied in other regions.





## 1. Introduction

The establishment of sensitive facilities such as nuclear power plants or dams have been raising the seismic risk to higher levels and entail the need for a profound understanding of the seismic hazard (e.g. Marano et al., 2010). Probably the most famous example is the destruction of the Fukushima nuclear power plant by tsunami waves caused by the 2011 $M_w = 9.0$ Tohoku-oki earthquake, which has been affecting an extensive region ever since. Identifying and characterising the regional seismic sources and their potential hazard is therefore fundamental for siting and designing of potential facilities, and for risk management. Additionally, in the case of infrastructures, the hazard also includes surface rupture in close proximity to the construction. The goals of this study are to define the regional main seismic sources, presuming that these are the sources that are likely to generate the most significant earthquakes in the near future, and to minimise the likelihood of surface rupture at the underlying infrastructure of sensitive facilities.

Despite the limited duration of the instrumental record, it constitutes one of the main direct evidence of fault activity in the current tectonic configuration. Probabilistic analyses of seismicity can constrain fault locations, kinematics and activity rates (e.g. Woo, 1996; Atkinson and Goda, 2011). Moreover, the Gutenberg-Richter empirical law allows assessing the frequency of medium to strong earthquakes by extrapolating low-magnitude earthquakes. Since surface ruptures are usually associated with M ≥ ~6.0 (Wells and Coppersmith, 1994; Stirling et al., 2002), the concentration of seismicity along faults highly suggests that surface ruptures occurred in the recent geological history. However, due to the scarcity of large earthquakes in the instrumental era, complementary information is required for further constraining the location of the main sources of significant earthquakes, and for characterising them. . This information can come from archaeological and paleo-seismological investigations, and from historical documents (e.g. Ambraseys, 2009; Agnon, 2014; Marco and Klinger, 2014). Geodetic measurements of relative displacements and velocities provide further crucial kinematic information (Baer et al., 1999; Hamiel et al., 2016; 2018a; 2018b).



Detailed geological investigation of faults can further extend the necessary
information, in particular for long-term activity. In terms of seismic hazard perspective,
faults that were active in the recent geological periods have a larger probability for future
faulting, compared with other faults. Field relations between faults and geological units,
as revealed in geological maps, can force constraints on the location, timing and the
amount of offset of the relevant faults. However, these evidences are limited to places
where faults have field relationships with young formations. Since the spatial distribution
of such formations can be limited, additional criteria are required for mapping potentially
hazardous faults.
In this paper we incorporate independent datasets to produce a variety of essential
products for seismic hazard evaluation, including surface rupture and ground motion. We
demonstrate it for the Israel region, a seismically-active zone mainly affected by the Dead
Sea Transform fault system (DST; Fig. 1). We first determine the main seismic sources in
Israel and its vicinity, focusing on faults that are likely to generate intermediate to large
earthquakes. Subsequently, we present the process utilised to determine and map faults
that constitute a potential hazard of surface rupture for sensitive facilities. We design the
criteria according to the likelihood of surface rupture along specific faults.

## 2. Tectonic settings

The continental crust in the region of Israel was formed during the Pan-African
orogeny of Late Precambrian age, and was later subjected to alternating periods of
sedimentation and erosion during the Paleozoic (Garfunkel, 1998). Continental breakup
and the establishment of passive margins along the Tethys-Mediterranean coast of the
Levant occurred during the Triassic-Jurassic time. Widespread carbonate platform
developed during the mid-Cretaceous. Since the Upper Cretaceous, the region was
subjected to WNW compression of the Syrian-Arc system, deforming the sedimentary
sequence into a series of asymmetric folds, strike-slip faults, and monoclines (Eyal and
Reches, 1983; Sagy et al, 2003). Regional uplift began from the end of the Eocene and
the area was intermittently exposed to erosional processes (Picard, 1965). The African-
Arabian plate broke along the suture of Gulf-of-Aden, Red Sea during the Miocene,



generating the Suez rift and the DST which separate the Sinai sub-plate from the African
and the Arab plates (Fig. 1). The Suez rift, however, has shown relatively minor signs of
deformation since the end of the Miocene (Garfunkel and Bartov, 1977; Joffe and
Garfunkel, 1987; Steckler et al., 1988), while the DST system remains the most active
tectonic feature in the area. In the Easternmost Mediterranean, the current plate boundary
deformation is taking place along the convergent Cyprian Arc (Fig. 1), where the
Anatolian plate overrides the plates of Africa and Sinai (e.g., Mckenzie, 1970).
The 1000-km DST is the largest fault system in the east-Mediterranean region (Fig.
1). Its northern section crosses northwest Syria in a N-S orientation; several recent large
earthquakes were attributed to this section during the past two millennia (Meghraoui et
al., 2003). The middle section of the DST is a restraining bend (LRB; Fig. 1),
characterised by transpression deformation (Quennell, 1959). The section is branched to a
few segments that transfer the main component of the strike-slip motion in Lebanon area
(Gomez et al., 2003; 2007). The Israel region is located along the southern section of the
DST but seismically it is also affected by the activity of the middle part.
The southern part of the DST (Fig. 1) is dominated by a sinistral motion of
approximately ~5 mm/yr, summing up to ~105-km of left-lateral displacement over a
period of 15-20 million years (e.g. Garfunkel, 1981; 2014). It is marked by a pronounced
5–25 km wide topographic valley, mostly with uplifted flanks, bordered by normal faults
that extend along the valley margins. The lateral motion occurs on longitudinal left-
stepping strike-slip and oblique-slip fault segments. The strike slip segments delimit a
string of en-echelon arranged rhomb-shaped narrow and deep releasing bends that are
associated with orthogonal separation of the transform flanks on the surface, which may
well extend beneath the crust (Garfunkel, 1981; Garfunkel and Ben-Avraham, 2001;
Wetzler et al., 2014). The seismic potential was clearly expressed by the 1995 $M_w = 7.2$
Nuweiba earthquake in the Gulf of Elat (Aqaba), the largest seismic event documented
instrumentally on the DST. Historical and prehistorical large earthquakes are also well
documented (e.g. Marco, 2008; Marco et al., 2005; Amit et al., 2002). The slip rates
along the DST vary between different fault segments and time resolutions, but converges
at about 4–5 mm/yr, approximately the same values obtained by GPS measurements
(Marco and Klinger, 2014; Hamiel et al., 2018a; 2018b). Deep-crust seismicity is



significant along the southern part of the DST in correlation with areas of low heat flow, particularly along the Dead Sea Basin, probably indicating a cool and brittle lower crust (Aldersons et al., 2003; Shalev et al., 2007; 2013).

The Sinai sub-plate south to Lebanon displays some amount of internal deformation expressed by a few fault systems, which are associated with Quaternary activity. The Carmel-Tirza Fault zone (CTF; Fig. 1) consists of a few normal and oblique fault segments generally striking NW-SE. The system is characterised by low heat flow and by relatively deep seismicity (Hofstetter et al., 1996; Shalev et al., 2013). The CTF divides the Israel-Sinai sub-plate into two tectonic domains (Neev et al., 1976; Sadeh et al., 2012) where the southern part is assumed to be relatively rigid, while northward, normal faults orientated E–W generate N-S extension expressed by graben and horst structures (Ron and Eyal, 1985).

## 3. Geological Database

The database of faults that were active in the recent geological history is mainly based on high-resolution geological maps. As of January 2019, 71 geological map sheets in the scale of 1:50,000 are available for this study, out of the 79 sheets required to cover the whole state of Israel (Fig. A1). The 1:200,000 geological map of Israel (Sneh et al., 1998) is utilised where 1:50,000 data are absent. Included also are faults defined as active or potentially active for the Israel Standard 413 "Design provisions for earthquake resistance of structures" (Sagy et al., 2013). In addition, some faults that have not been mapped (or not updated yet) crossing Quaternary units in the geological maps, are marked here as Quaternary faults based on evidence presented in scientific publications, reports, and theses (see Table A1).

The establishment of Quaternary formation database (Table A2), to constrain fault activity in this study is complicated due to poorly constrained geochronology of some of the formations. In some cases the age uncertainty is in the order of millions of years. Moreover, the boundary Pleistocene-Pliocene (Neogene-Quaternary) was shifted in 2009, from ~1.8Ma to ~2.6Ma. Thus, some formations that had previously been assigned Pliocene age became part of the Pleistocene. Therefore, geological periods attributed to



some formations, mentioned in pre-2009 publications, might mislead. Many stratigraphic
charts of the pre-2009 geological maps are outdated. Furthermore, as recent research
provides better geochronological constraints, the most up-to-date information is required
in order to correctly select Quaternary formations. In Appendix 1 (Table A1) we present
references to Quaternary faults that cannot be directly deduced from the geological maps.
Beside the surface traces of mapped faults, offshore and subsurface continuation of
faults, as well as faults extending beyond the Israeli borders were added to the database
(Table A3). The latter are limited to the extensions of mapped faults that are within
Israel, and/or the main DST segments. The criteria for selecting these faults are discussed
in section 6.

## 4. Seismological analysis

We analyse the spatial distribution of seismic events in order to reveal the regional
seismic pattern, which helps to define the main seismic sources and develop an
independent criterion for Quaternary active faults. In order to define the seismicity-based
criterion, we designe seismic criteria that are based on the distribution of two parameters:
the *Earthquake Kernel Density* and the *Seismic Moment Kernel Density*. We demonstrate
the methodology and then present the results below.

### 4.1 Dataset

We use an earthquake catalogue from 1.1.1983 until 31.8.2017 within $28°N - 34°N$
and $33°E - 37°E$, recorded by ~140 stations whose distribution has changed in time and
space. Most of the data are from the Israel Seismic Network (ISN), the Comprehensive
Nuclear Test-Ban Treaty (CTBT), and the Cooperating National Facility (CNF). Some
additional data were incorporated from other regional networks: GE, GEOFON global
network of Deutsches GeoForschungsZentrum, Potsdam (GFZ), Jordanian Seismic
Observatory (JSO), and the seismic network of Cyprus (CQ). These earthquakes, which
have been monitored by the Seismological Division of the Geophysical Institute of Israel,
comprise a catalogue of ~17,600 earthquakes. They were relocated (Fig. 2) to generate a





new catalogue with more precise locations of hypocentres (Wetzler and Kurzon 2016).
As part of the relocation process, ~900 earthquakes were excluded for various reasons,
e.g., events that were recorded by less than 4 stations; large location errors (including the
$M_d = 5.8$ 1993 event in the Gulf of Elat). Before 1983 the locations are less reliable.
Hence, the relocated catalogue consists of ~16,700 events of $0.1 \leq M \leq 7.2$ (Fig. 2).
Earthquakes with unknown magnitudes received a default value of M = 0.1. The
magnitude and the location of the $M_w = 7.2$ 1995 Nuweiba earthquake were fixed
according to Hofstetter et al. (2003).
In order to assess the applicability of the following seismic processing and analysis,
we define the network coverage area as the zone in which the hypocentres are relatively
well-constrained. This is examined and determined here as the polygon that covers all
seismic stations that recorded at least 350 arrivals, and consists of the smallest number of
polygon-sides that link between the stations (Fig. A2 in Appendix 2).

**4.2 Spatial data processing**
In order to quantitatively characterise the regional seismicity and associate the
earthquakes with mapped faults we examine two parameters: a) *earthquake kernel*
*density* and b) *seismic moment* ($M_0$) *kernel density*. Both parameters are obtained through
the following spatial data processing. A regional scan is carried out in a 0.5-km interval
2D grid, in the horizontal coordinates. For each grid point, both parameters are calculated
for the events within a 6-km distance of the grid point. The parameters are calculated
based on the kernel density estimation as an approach to obtain the spatial distribution
through a probability density function, using the distance to weight each event from a
reference point (each grid point). The weighting can be illustrated as many circles of up
to 6-km radius that surround a common centre (every grid point). The circle shape
prevents any directional bias.
The 6-km radius from each grid-point, and the Gaussian function and its standard
deviation of 2 (for the kernel estimation), were tuned and chosen to: a) capture different
seismic patches along active faults; b) be significantly larger than the location horizontal
median error (~1.2 km; Wetzler and Kurzon, 2016); c) assign higher weight to events





closer to the evaluated grid-point; d) include as many events as possible for achieving
statistical significance at each of the grid-points.

The *earthquake kernel density* parameter, $\rho_{Nk}$, is calculated by counting all the

weighted events within a 6-km radius from each grid point, dividing their sum by the
sampler area ($\pi r^2$) and normalising by the duration of the earthquake catalogue:

$$\rho_{Nk} = \frac{\sum_{n=1}^{N} e^{-\frac{d(n)^2}{2\sigma^2}}}{T\pi r^2}$$    (1)

where $N$ is the total number of events within the radius $r$, $d(n)$ is the distance between an
event $n$ and the circle centre; $\sigma$ is the standard deviation of the Gaussian function, and $T$
is the duration of the earthquake catalogue. Units are $[events/km^2/yr]$.

The $M_0$ *kernel density* parameter, $\rho_{M0k}$, is obtained by first calculating the seismic

moment released by each event separately, using the empirical relation between $M_0$ and
$M_L$, as obtained by Shapira and Hofstetter (1993) after converting units from *dyne-cm* to
*N-m*:

$$log[M_0] = 10 + 1.3M_L$$    (2)

Secondly, each amount of energy is weighted according to the distance of the
corresponding event from the circle centre (like the calculation of the *earthquake kernel*
*density*). Then, we sum the weighted-$M_0$ released from all the events within a 6-km
radius, divide the sum by the circle area ($\pi r^2$) and normalise by the duration of the
catalogue:

$$\rho_{M0k} = \frac{\sum_{n=1}^{N} M_0(n) e^{-\frac{d(n)^2}{2\sigma^2}}}{T\pi r^2}$$    (3)

where $N$ is the total number of events within the radius $r$, $M_0(n)$ is the seismic moment
released from an event $n$ according to Eq. 2, $d(n)$ is the distance between an event $n$ and
the circle centre, $\sigma$ is the standard deviation of the Gaussian function, and $T$ is the
duration of the earthquake catalogue; units are $[joule/km^2/yr]$.






### 4.3 Distribution maps of the spatial processing parameters

#### 4.3.1. Earthquake Kernel Density

The *earthquake kernel density* (Fig. 3) captures the main active tectonic sources and seismic patches, according to ~35 years of instrumental seismicity. As expected, most of the earthquakes are concentrated along the main fault zone of the DST, and to a lesser extent along the CTF, including its offshore continuation in the Mediterranean Sea. In the southwest, seismicity is observed in the area of the Gulf of Suez. Small patches appear in different spots, mainly west of the DST, raising the issue of the detectability of the network east of it. We note that the International Seismological Centre catalogue reveals large portion of events recorded east of the DST as well (Palano et al., 2013). The most prominent zone of seismicity that is not associated with known active tectonic feature is northwest of the Gulf of Elat.

A more detailed scan of the seismicity from south shows that the prominent patches of seismicity along the DST are located in the Gulf of Elat, the Arava valley, and the Dead Sea Basin. Northwards, seismicity becomes more distributed, reflecting the intersection between the DST and the CTF (Fig. 1). North of the intersection, the Jordan valley segment of the DST is sparse with seismicity. However, further north, dominant seismicity patches are seen in the Sea of Galilee, and in the Hula valley. Northwest of the Hula valley, another zone of intense seismicity is captured, which might be associated with faults related to the Roum fault, west of the LBR (Meirova and Hofstetter, 2013).

#### 4.3.2. Seismic moment kernel density

The distribution of the average annual moment density released from all earthquakes, assuming them as point sources, is shown in figure 4. Since the amount of energy released by each earthquake differs significantly according to its magnitude, this parameter is presented on a logarithmic scale. Overall, the *Mo kernel density* distribution emphasises the seismic activity along the DST, with similarity to the *earthquake kernel density* distribution (Fig. 3). Still, the distribution is less smooth due to single events differing significantly from each other in their corresponding Mo release.





The Gulf of Elat includes the largest event recorded in the catalogue, the $M_W = 7.2$
1995 Nuweiba earthquake (Hofstetter et al., 2003), two order of magnitudes larger than
the second-largest event ($M_d = 5.6$), hence the significantly higher values in its vicinity.
The spatial distribution of the *Mo kernel density* reveals a wide zone of deformation
surrounding the gulf flanks, much wider than the relatively narrow gulf. This can be
partially explained by the poorly-constrained epicentre locations, far away from the
network coverage (Fig. A2). The *seismic moment kernel density* reflects strongly the most
significant events that occurred in the past 35 years; among them are the $M_w = 5.1$ 2004
event in the Dead Sea (Hofstetter et al., 2008), and the $M_d = 5.3$ 1984 event associated
with the CTF. In contrast with the distribution of the *earthquake kernel density*, the *Mo*
*kernel density* does not reflect seismic swarms, unless they consist of high magnitudes.
This contrast is predominant in the Sea of Galilee, which contains high *earthquake kernel*
*density* (Fig. 3) but is less significant in the *seismic moment kernel density* (Fig. 4).

## 278    5. The main seismic sources

Figures 3 and 4 show a strip of dense seismic events and moment release along the
DST and its main branches. We now combine these data with geologic, geodetic and
paleoseismologic measurements to generate the main seismic sources map, which
displays regional faults that demonstrates slip rates inferred here as $\geq 0.5$mm/yr during
the Holocene. Tectonic and geometric characteristics (i.e., segment length & orientation)
are also considered. We define the main seismic sources as faults that are likely to
generate significant earthquakes (M ≥ 6.0), which can impact Israel and constitute
potential sources for different sorts of damages (i.e., ground motion and acceleration,
landslides, liquefactions and tsunamis). These faults and their map (Fig. 5) are essential
for seismotectonic modelling of Israel, Probabilistic Seismic Hazard Analysis (PSHA)
and eventually for generating ground motion maps. Below, we define two subgroups of
faults divided by their tectonic characteristics and their slip rates. Off-shore inferred
continuations of the main faults are also presented (dashed lines in Fig. 5).

### 292    5.1 Potential sources for large earthquakes



This category (solid black lines in Fig. 5) includes the main sinistral and oblique fault
segments of the DST in the region. According to paleoseismic and/or geodetic
investigations (Table 5), these faults are associated with Holocene slip rates of 1 mm/yr <
$V_S$ < 5 mm/yr, where $V_S$ is the average sinistral slip component accommodated by these
faults. Equally important, all the faults in this category are relatively long with a
preferable slip orientation according to the present stress field (Jaeger et al., 2007). Our
database (Fig. 5) includes fault segments from this subgroup which are located up to 150-
km away from Israel. As noted in Sec. 4, the only recorded large earthquake, the 7.2 $M_W$
Nuweiba earthquake occurred on the Aragonese Fault and was associated with mean slip
of 1.4–3 m (Baer et al., 1999).
South to Lebanon, geodetic measurements show ~ 4–5 mm/yr sinistral slip (Hamiel et
al., 2016; 2018a; 2018b; Masson, 2015). Faulting in Lebanon is partitioned to a few
branches (Fig. 3) and the specific rates are less constrained. While the Yammuneh and
the Serghaya faults can undoubtedly be considered as independent sources for significant
earthquakes, the status of the shorter, Rachaiya and Roum fault branches are less clear.
Nevertheless, according to the present state of information (see for example, Nemer and
Meghraoui (2006)), we cannot rule them out and they remain part of this group.
Previous analyses of maximum earthquake magnitude based on historical earthquakes
or on background seismicity predicted magnitudes of ≤ 7.8 $M_w$ for the largest segments
(e.g., Stevens and Avouac., 2017; Klinger et al., 2015; Hamiel et al., 2018a).

### 5.2. Potential sources for intermediate earthquakes

This category (pale blue lines in Fig. 5) consists of fault zones with lengths of several
dozen kilometres that are associated with the DST, and display estimated slip rates of 0.5
mm/yr ≤ $V_S$ ≤ 1 mm/yr (Table 6).
This subgroup includes the fault zone in the western and eastern margins of the Dead
Sea; the marginal faults of the Hula basin and the CTF (Fig. 5). The partitioning of the
slip rate across parallel segments in any given fault zone is usually below the geodetic





measurement (or the information) resolution. Therefore, the segments of this category in
Figure 5 are representative, but not necessarily the most active within a given system.

Due to the lack of reliable historical and paleo-seismological evidences, the
evaluation of maximum possible magnitude on these faults is usually hard and requires
several assumptions. First, we consider here local rupture on a segment from a given
system and disregard a rupture of the entire system as part of an extremely large
earthquake on the main strike-slip faults (such a rupture is discussed in Sec. 5.1). In
addition, we assume that the longest possible subsurface rupture length is similar to the
length of the segment's surface trace. For example, the Carmel Fault, the northern fault in
the CTF is up to 40-km length (on and off shore). According to some published scaling
relationships, rupturing along its entire length can be associated with up to ~ 7 $M_w$
earthquakes (Wells and Coppersmith, 1994; Stirling et al., 2013). However, we assume
again that such magnitudes must be interconnected with an earthquake along a much
larger DST segment, and not confined to a local fault (Agnon, 2014). We therefore
assume a maximum rupture length of ~10–20 km along faults from this subgroup and
correspondingly to maximum magnitudes of $6.0 < M_w < 6.5$ (Wells and Coppersmith,
1994). The data on the Elat Fault is based only on evidence from its northern edge while
the rates at its offshore parts are less constrained. Shaked et al. (2004) inferred a
catastrophic event at 2.3ka on the Elat Fault.
Large earthquakes along the Cyprian Arc (Fig. 1) can also generate tsunamis that
might affect the coastline of Israel (Salamon et al., 2000). This source is not analysed and
mapped here, but should be taken into account in regional seismotectonic models.

## 6. Capable faults

### 6.1 Framework and principles

The hazard of surface rupture is defined as the likelihood of an earthquake that will
rupture the surface within a certain time window. This likelihood is based on knowledge
about the past and present fault kinematics and dynamics. The determination of the
relevant time reference for young faulting is usually dictated by different constrains and



applications. In the United States, faults are commonly considered to be active for planning constructions if they have ruptured the surface at least once in the past 10ka. However, regional conditions, such as sedimentary cover or available age dating of pertinent geological units can affect this determination. For example, faults that are defined as "Active" in the "Design Provisions for Earthquake Resistance of Structures" in Israel are those that ruptured the surface in the past 13ka (Heimann, 2002). This is the age of the top of the lake formation that covers significant parts of the Dead Sea valleys.

The time reference for special constructions such as dams and nuclear power plants is usually much longer, because the possible damage to the construction has severe regional implications. According to the International Atomic Energy Agency (IAEA) Safety Fundamentals (2010), capable faults are these with evidence for displacement since thousands or millions of years, depending on the region activity. Here, the Quaternary period is selected as the time reference for sensitive facilities due to two main reasons: a) we assume that faults that were active during the present regional stress regime (Zoback, 1992) are more likely to activate in the near future. The regional stress state within the Quaternary period is represents well the current stress field (Eyal and Reches, 1983; Hofstetter et al., 2007; Garfunkel, 2011; Palano et al., 2013). We note that "regional stress field" (Zoback, 1992) as a criterion for active faulting is closely related to the "tectonic regime" suggested by Galadini (2012). b) Quaternary geological units are mostly well defined in the region.

The primary and secondary criteria for sorting the faults are listed in a descending order of categorisation, meaning that faults are initially examined according to the first criterion, and only if they do not match it, they are examined according to the second criterion, and so on.

Finally, in regions where Quaternary cover is absent, we utilise a seismological criterion (Fig. 6), based on the assumption faults that are associated with seismically active subzones are more likely to have ruptured the surface in the Quaternary compared to others.



### 6.2 Primary criteria

1. Main strike-slip faults of the DST: identified here as main sources for large regional earthquakes (Fig. 7).

2. **Faults with direct evidence of Quaternary activity:** faults that have been mapped offsetting Quaternary formations or that have been interpreted by scientific publications (Table A2) to rupture the earth's surface at least once since the Quaternary. This criterion is mainly related to zones covered by Quaternary units.

### 6.3 Secondary criteria

Faults that have no field relationship with Quaternary formations consequently show no direct evidence for Quaternary faulting. We therefore designed the next criteria under the rationale that they expand the database with faults that reasonably have been active since the Quaternary, based on the following three sub-criteria:

1. First order branches and the marginal faults of the DST

   **a)** First order branches of faults that are mapped following the primary criteria. A fault branch is defined here as splitting at an acute angle from another fault. The throw direction of the fault and its branches are also taken into account.
   **b)** Faults that bound the DST basins, separating Quaternary formations from older rocks and are associated with a sharp topographic boundary of at least 100 meters.
   **c)** Faults that emerge from Quaternary sediments that infill the DST valleys and are likely to branch off of the main DST segments.

2. Faults associated with recent seismicity

   it is challenging to match the faults and recent seismicity and assume they ruptured the surface at least once since the beginning of the Quaternary because there are thousands of mapped faults, high-resolution geophysical data about the fault structures in depth are scarce, and the hypocentres' location uncertainties are large. In order to define the seismicity-based criterion, we create polygons for each of the parameters. The polygons are defined by a threshold value, so that each of them is





the smallest to cover the most active tectonic in the region, continuously in this case,
the DST; excluding the relatively silent northern section of the Jordan Valley
segment (I in Fig. 6). Therefore, the overlap area (Fig. 6) of the two polygons
consists of at least the minimum level of both *seismic moment kernel density* and
*earthquake kernel density*, along the DST in the Israel region. Hence, if a fault is
within the overlap area, it means that it is associated with at least a minimum level of
seismicity along the most active tectonic feature, and thus it is likely to be
seismogenic. We further assume a relation between a fault mapped surface trace and
a possible past surface rupture, in order to select the most prominent faults.
Considering scaling relations between fault dimensions and source parameters, faults
that contain surface traces of at least 6-km (corresponding to $M_W \geq 6.0$ earthquakes;
Wells and Coppersmith, 1994; Stirling et al., 2002; Mai and Beroza, 2000) within the
'overlap area' are assumed here as Quaternary faults.
3.  Subsurface faults
Subsurface and offshore continuation of the main DST strike-slip segments, and a
few other faults with published details for both their subsurface extension and their
Quaternary activity are marked (the majority are in Fig. 5). In addition, we map other
faults that offset dated Quaternary units, with well-constrained near-surface location
inferred from high-resolution seismic data. We exclude subsurface faults when their
exact location and activity period less constrained. Fault segments that were mapped
as concealed (mostly by thin alluvium) in the 1:50,000 maps and are the continuation
of Quaternary faults are marked as ordinary surface traces.

## 432  **7. Discussion**

Regions with intermediate seismicity rates present a challenge for hazard evaluation;
while the hazard is perceptible, the seismic data is sparse comparing to very active zones.
Taking into the account that the earthquake phenomenon is a stochastic process and its
predictability is limited, we develop a methodology that takes advantage of incorporating
interdisciplinary information with statistical analyses for seismic hazard evaluation. We
delineate the distribution of the density of earthquakes and of the seismic moment release





by analysing recorded seismicity and applying statistic-based data processing (Figs. 3, 4).
However, instrumental seismological data is practically limited, and the precision of the
results depends on the amount and the quality of the data, regardless of the specific
statistical method. This gap is closed by geodetic measurements, paleo-seismology and
historical information.
Throughout the capable fault map (Fig. 7), the information about the seismic intervals
of most of the faults is poor compared with these of the DST main strike-slip faults.
Faults of different categories are distributed in the same areas: these that show direct
evidence of Quaternary faulting, and those that fit seismo-tectonic criteria. For example,
branches of the DST main segments that do not cross Quaternary sediments, are marked
based on tectonic rationale. Moreover, although faults are marked by hierarchical criteria,
in many cases the different categories complement each other rather than show hierarchy
of the activity level. Accordingly, the distribution of the different faults is rather
homogeneous throughout the map (Fig. 7). This includes faults marked based on the
seismicity-based criterion. The Quaternary faults are superimposed on the seismicity
polygons of this criterion (Fig. A3) and reveal that many the majority of the faults, which
are mapped based on the geological criteria, could have enter the map also by the
seismological criterion (ignoring its 6-km fault length limitation). Thus, the correlation
between the recorded seismicity and the Quaternary faults support the design of the
seismicity-based criterion. On the other hand, we do not define faults that constitute a
mechanical potential for slip (for example, conjugate fault sets) as capable, unless further
geological or seismological evidence for Quaternary activity is existed. Such a
mechanical criterion, however, should be considered and re-evaluated during the specific
siting stage.
While most of the seismic activity follows the DST, some areas along it are associated
with very sparse seismicity (Fig. 6). At the northern section of the Jordan Valley
segment, section I is the least active part of the DST during the last ~35 years. Geodetic
analysis demonstrates that this section creeps at a rate of approximately half of the total
plate motion (Hamiel et al., 2016). This creep, together with potential partitioning of the
activity to the CTF, might cause the relative reduction of earthquakes in section I (Fig. 6).
Sections II and III, at the middle and the northern sections of the Arava segment, are also



associated with sparse seismicity, but to a lesser extent. With no indication for creep, the reduction of seismicity might be attributed to local locking of the main fault or to the influence of other structures in fault junctions (e.g. WSW-ENE orientated faults of the Sinai-Negev shear belt (Bartov, 1974)). Further research of these zones is required for better understanding the local variation of the seismic patterns.

## 8. Conclusions

1. Mapping and characterising faults that pose seismic hazard require generating interdisciplinary regional database and developing hierarchical seismo-tectonic criteria. With respect to the specific dictated requirements, faults that are potential sources for the far-field and for the near-field (i.e., surface rupture) hazards should be analysed by different criteria; both represent seismic hazard of significant earthquakes but within different time frames.

2. The regional main seismic sources are primarily defined by the recent slip rates. Geologic and geodetic slip rates, as well as long historical record and high-resolution mapping enable reliable definition of faults that are likely to generate large earthquakes. All the main seismic sources in the Israel region (Fig. 5) are related to the DST activity.

3. The time reference for local planning of special constructions such as dams and nuclear power plants is usually long, because the possible damage to the construction has severe regional implications. We selected the Quaternary period as the relevant time frame for capable faults in the region of Israel. While this time frame (2.6 Ma) is longer than the previous for defining capable faults for a potential local nuclear power plant (IEC and WLA, 2002), it is justified by considering the regional stress field, the regional stratigraphic configurations and the criteria that focus on surface rupture rather than general fault movements. We conclude that tectonic and stratigraphic conditions, as well as the accessibility of geologic maps and their resolutions, should be taken into account for defining the time frame for capable faults.

4. We design a seismicity-based criterion that is based on the distribution of two parameters: the *Earthquake Kernel Density* and the *Seismic Moment Kernel Density*. The



success of this selection is further reinforced by the match between the geological-categorised faults and the seismicity criterion (Fig. A3).

5. Beyond planning of special constructions, the developed database and the maps that are generated and presented here constitute further applications for planning and research. The regional main seismic sources map (Fig. 5) is fundamental for seismotectonic modelling and eventually for generating ground motion prediction maps (e.g. by PSHA) that include essential information for construction planning, such as peak ground acceleration. The capable fault database and the related maps (Figs. 2-4, 6-7) lay the foundation for further study of the regional Quaternary faulting and tectonics in the Israel region. Furthermore, the methodology, which is based on categorisation and sub-categorisation by seismo-tectonic hierarchic criteria, enables differentiation of hazard potential and can be applied in other regions around the world.

## Acknowledgments

We thank the following people for their collaboration and assistance: R. Amit, Y. Avni, Y. Bartov, Z. Ben-Avraham, G. Baer, M. Beyth, A. Borshevsky, R. Calvo, Y. Eyal, Z. Garfunkel, H. Ginat, Z. Gvirtzman, Y. Hamiel, S. Hoyland, S. Ilani, R. Kamai, W. Lettis, T. Levi, D. Mor, C. Netzer, P. Nuriel, Y. Sagy, A. Salomon, A. Sneh, R. Weinberger, E. Zilberman.

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





*Table 1: Main strike-slip faults: average slip rate details*

| Fault | Strike-slip [mm/yr] | Data | Period | Reference |
|---|---|---|---|---|
| Aragonese [ARF] | ~5* | GPS | Recent | Baer et al. 1999; Hamiel et al., 2018a |
| Arava [AF] | ~4.9# | GPS | Recent | Masson et al., 2015 |
| Evrona [EF] | 5.0±0.8# | GPS | Recent | Hamiel et al., 2018a |
| Jericho [JF] | 4.8±0.7#! | GPS | Recent | Hamiel et al., 2018b |
| Jordan Valley [JVF] (central) | ~5# | Geology | ~25ka | Ferry et al., 2011 |
| Jordan Valley (South to Sea of Galilee) | 4.1±0.8#& | GPS | Recent | Hamiel et al., 2016 |
| Jordan Gorge | 4.1±0.8# <br> ~3# <br> ~2.6# | GPS <br> Geology <br> Archaeology | Recent <br> ~5ka <br> ~3ka | Hamiel et al., 2016 <br> Marco et al., 2005 <br> Ellenblum et al., 2015 |
| Lebanon Restraining Bend (LRB) | 3.8±0.3* | GPS | Recent | Gomez et al., 2007 |
| Qiryat Shemona | 3.9±0.3*! | GPS | Recent | Gomez et al., 2007 |
| Roum | 0.86–1.05# | Geology | Holocene | Nemer and Meghraoui, 2006 |
| Serghaya | 1.4±0.2# | Geology | Holocene | Gomez et al., 2003 |
| Yammuneh (LRB – northern part) | 2.8±0.5 | GPS | Recent | Gomez et al., 2003; 2007 |
| Yammuneh (north of LRB) | 6.9±0.1# <br> 4.2±0.3* | Geology <br> GPS | 2ka <br> Recent | Meghraoui et al., 2003 <br> Gomez et al., 2007 |

\# Geodetic or geological measurements on a specific segment.

! 0.8 mm/yr of extension normal to the fault

\* According to geodetic-based model

& Partially creeping



*Table 2. Marginal faults and branches with integrated slip or subsidence of ~0.5 mm/yr*
*≤ VS ≤ ~1 mm/yr and references*

| Fault | Slip rate [mm/yr] | Data | Period | Reference |
|---|---|---|---|---|
| Dead Sea basin marginal faults | ≥1 Based on basin subsidence rates | Geology Geophysics | Pleistocene-Holocene | Torfstein et al., 2009; ten Brink and Flores, 2012; Bartov and Sagy, 2004 |
| Carmel-Tirza-Izrael fault zone [CTF] | 0.9±0.45 total slip rate (0.7±0.45 lateral; 0.6±0.45 extension) | GPS | Recent | Sadeh et al., 2012 |
| Carmel | < 0.5 | Geology | 200ka | Zilberman et al., 2011 |
| Hula western border | > 0.4 Based on basin subsidence rates | Geology Geophysics | ~1 Ma | Schattner and Weinberger, 2008 |
| Elat | ? | Geology | Holocene | Amit et al., 2002; Porat et al., 1996; Shaked et al., 2004 |




**Figure captions**
**Figure 1: Plate configuration in the Eastern Mediterranean. Arrows show relative motion.**
**SR-Suez Rift; GEA: Gulf of Elat/Aqaba. DST-Dead Sea Transform fault system; CTF-**
**Carmel Tirza Fault zone; LRB-Lebanon Restraining Bend; CA- Cyprian Arc.**
**Figure 2: Epicentres in Israel and surrounding areas between the years 1983-2017, based on**
**the relocated earthquake catalogue. Circle size and colours indicate the magnitude. Black**
**lines represent the main fault segments of the DST and the CTF. The background for this**
**figure and the followings is based on Farr et al., (2007).**
**Figure 3: The earthquake kernel density distribution, according to the relocated catalogue.**
**Colours and corresponding numbers indicate the value in [events/km$^2$/yr].**
**Figure 4: The seismic moment kernel density distribution, according to the *relocated***
***catalogue. Colours and corresponding numbers indicate the value in* $log[joule/km^2/yr]$.**
**Figure 5: The main seismic sources in Israel and adjacent areas. Colours indicate the two**
**categories of faults according to the criteria. Inferred subsurface faults are marked by**
**dashed lines. Abbreviations are for the DST main strike-slip segments, its main branches**
**and marginal faults. Numbers indicate geodetic slip rates [mm/yr] for strike-slip**
**components, according to recent studies (Tables 1, 2).**
**Figure 6. The seismicity polygons: earthquake density of values > ~0.001[events/km$^2$/yr] and**
**Mo density of values > ~9.5 $log[joule/km^2/yr]$; the product is the overlap polygon (in**
**brown).**
**Figure 7. Quaternary fault map of Israel. Colours indicate the corresponding criterion for**
**each fault. Inferred subsurface faults are marked by dashed lines. Abbreviations are for the**
**main strike-slip segments of the DST.**








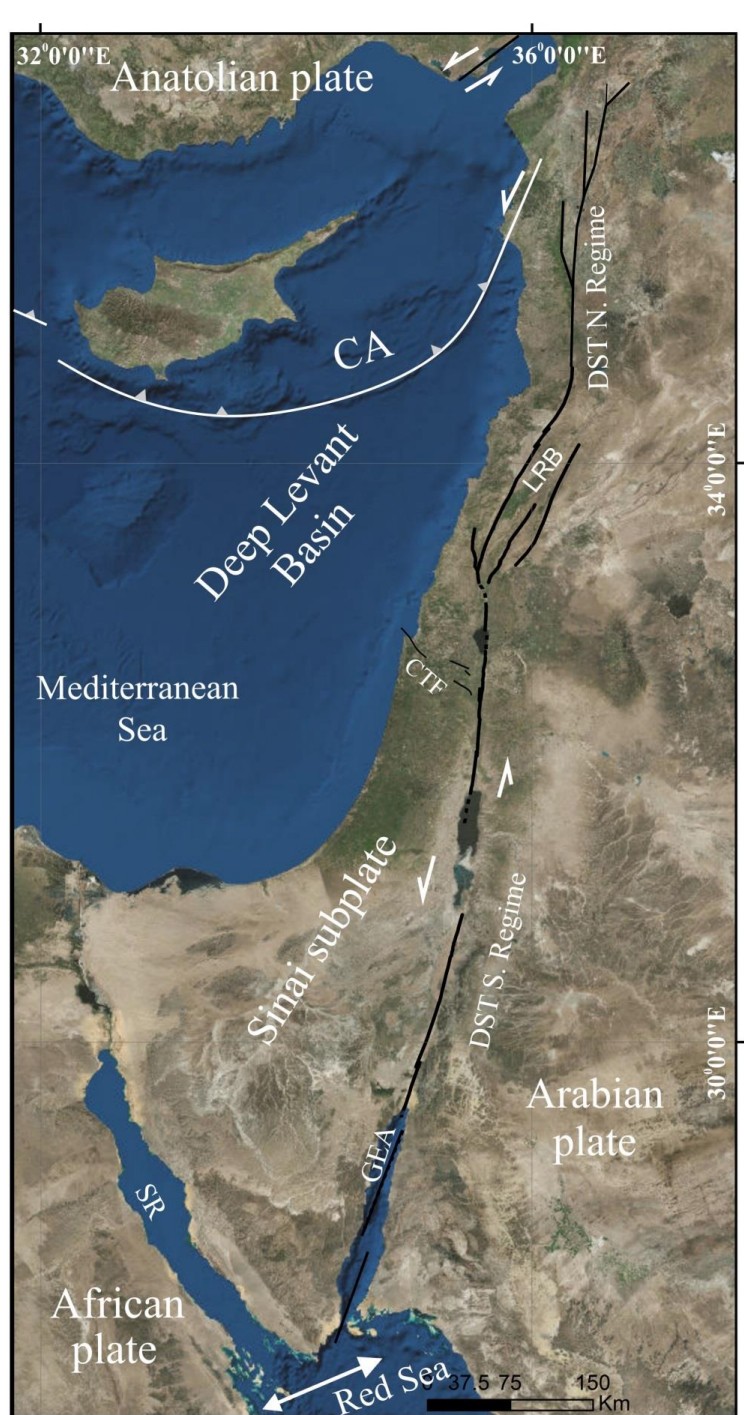


**Figure 1**






**Figure 2**









**Figure 3**





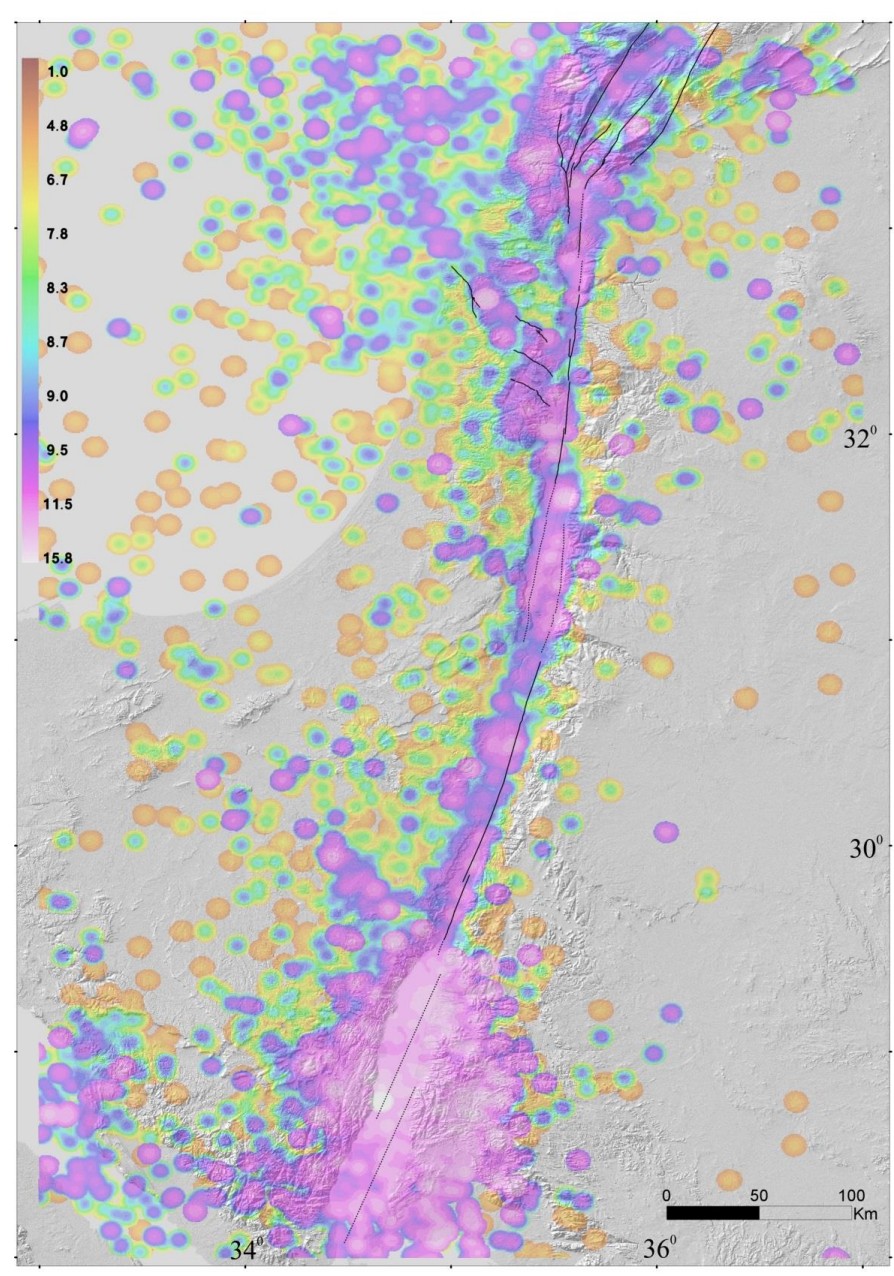

**Figure 4**






**Figure 5**






**Figure 6**







**Figure 7**





**Appendix 1**

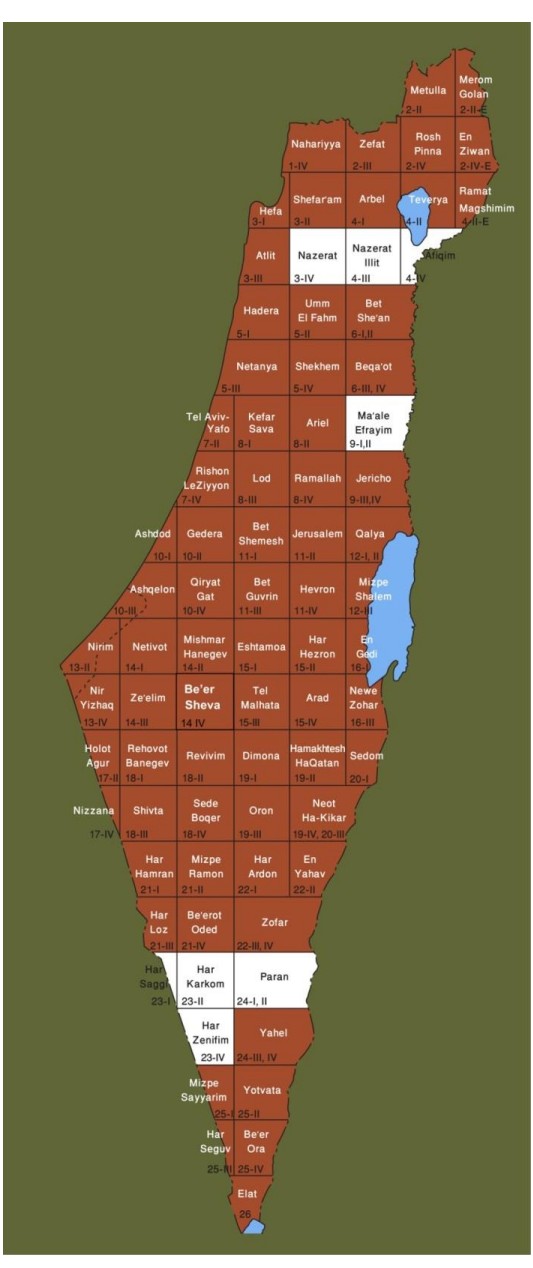


Figure A1. Locations of the 1:50,000 geological map
sheets used for the present map (as of August 2018).
Brown: locations of published 1:50,000 sheets.
White: unpublished sheets.



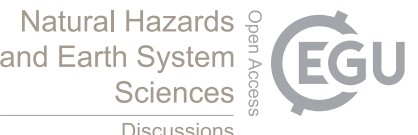

*Table A1: References for faults and fault segments that have been marked based on*
*papers, reports, and theses. Faults are listed in table 3 if their latest mapping is not*
*updated yet in the 1:50,000 sheets (as of 2018), or if their definition as Quaternary*
*faults cannot be directly deduced from the geological maps. Fault names are mainly*
*according to the references.*

| Area | Name of fault / group of faults or segments | References |
|---|---|---|
| Southern Israel | Arif-Bator | Zilberman et al., 1996; Avni, 1998 |
| | Gerofit | Ginat, 1997 |
| | Gevaot Ziya | Avni, 1998 |
| | Halamish line | Avni, 1998 |
| | Har Seguv | Avni, 1998 |
| | Hiyyon | Ginat, 1997 |
| | Katzra | Avni, 1998 |
| | Milhan | Ginat, 1997 |
| | Mitzpe Sayarim | Avni, 1998 |
| | Noza | Ginat, 1997 |
| | Ovda | Avni, 1998 |
| | Paran | Zilberman, 1985; Avni, 1998; Calvo et al., 1998; Calvo, 2002 |
| | Yotam | Wieler et al., 2017 |
| | Zhiha | Avni, 1998 |
| | Zin | Enzel et al., 1988; IEC and WLA, 2002; Avni and Zilberman, 2007 |
| | Znifim – Zihor – Barak | Ginat, 1997 |
| | Zofar | Calvo, 2002 |
| Central Israel and Dead Sea area | Jericho | Sagy and Nahmias, 2011 |
| | Masada Plain | Bartov et al., 2006 |
| | Modi'in | Buchbinder and Sneh, 1984 |
| | Nahal Darga (east) | Enzel et al., 2000 |
| | Nahal Kidron (east) | Sagy and Nahmias, 2011 |
| Northern Israel | Ahihud | Kafri and Ecker, 1964; Zilberman et al., 2011 |
| | Beit Qeshet (western part) | Zilberman et al., 2009 |
| | Ha'on | Katz et al., 2009 |
| | Hilazon | Kafri and Ecker, 1964; Zilberman et al., 2008 |
| | Kabul | Kafri and Ecker, 1964; Zilberman et al., 2008 |
| | Nahef East Fault | Mitchell et al., 2001 |
| | Nesher | Zilberman et al., 2006; 2008 |
| | Tiberias | Marco et al., 2003 |


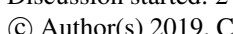



*Table A2: List of geological formations and units used for the QFMI Geologic and*
*geomorphic descriptions that appear in 1:50,000 geological maps for Quaternary*
*deposits.*

| Formations | Local sedimentary units | Local volcanic units | Other units* |
|---|---|---|---|
| Ahuzam Fm. (Cgl.) | Amora Salt | Avital Tuff | Alluvium |
| Arava Fm. | Betlehem Cgl. | Bene Yehuda Scoria | Beach rocks & reefs |
| Amora Fm. | Biq`at Uvda Cgl. | Brekhat Ram Tuff | Calcareous sandstone (kurkar) |
| Ashmura Fm. | Edom facias | Dalton Basalt | Colluvium |
| Garof Fm. | Egel Cgl. | Dalton Scoria & Tuff | Dune sand, Sand sheets, Red sands |
| Gesher Bnot Ya'aqov Fm. | En Awwazim Cgl. | Dalwe flows | Loess, fluvial & eolian |
| Hazor & Gadot Fms. | En Feshha Cgl. | En Awwazim flow | Gypsum |
| Lisan Fm. | Giv'at Oz Cgl. | En Zivan Basalt flows | Lake sediments |
| Malaha Fm. | Karbolet caprock | Golan Basalt flows (Muweissa and En Zivan flows) | Loam (hamra) |
| Mazar Fm. | Lot caprock | Hazbani Basalt flows | Neogene-Quaternary conglomerate units, Terrace cgl. |
| Nevatim Fm. | Mahanayim Marl | Keramim Basalt | Playa |
| Ortal Fm. | Mearat Sedom caprock | Meshki Basalt flows | Recent fan |
| Pleshet Fm. | Nahshon Cgl. | Muweisse Basalt flows | Soil |
| Samra Fm. | Ramat Gerofit Cgl. | Neogene Basalts | Tufa, travertine |
| Sede Zin Fm. | Ravid Cgl. | Raqad Basalt | Unnamed clastic unit |
| Seif Fm. | Ruhama Loess & sand | Sa'ar Basalt flows | |
| Ye'elim Fm. | Sabkha soil | Shievan Scoria | |
| Ze'elim Fm. | Si'on Cgl. | Yarda/Ruman Basalt flows | |
| Zehiha Fm. | Wadi Malih Cgl. | Yarmouk Basalt | |
| | | Yehudiyya & Dalwe Basalt flows | |







*Table A3: References for faults located beyond Israel borders and/or subsurface faults*

| Geographic area | Reference |
|---|---|
| Gulf of Elat | Ben-Avraham, 1985; Hartman et al., 2014; |
| Arava valley | Calvo, 2002; Le Béon et al., 2012; Sneh and Weinberger, 2014 |
| Sinai peninsula | Sneh and Weinberger, 2014 |
| North-western Negev | Eyal et al., 1992 |
| Dead Sea basin | Ben-Avraham and Schubert, 2006; Sneh and Weinberger, 2014 |
| Jordan valley | Ferry et al., 2007; Sneh and Weinberger, 2014 |
| Gilboa fault (western part) | Sneh and Weinberger, 2014 |
| Carmel fault (eastern part) | Sneh and Weinberger, 2014 |
| Carmel fault (western part) | Schattner and Ben-Avraham, 2007 |
| Zvulun Valley | Sagy and Gvirtzman, 2009 |
| Sea of Galilee | Hurwitz et al., 2002; Reznikov et al., 2004; Eppelbaum et al., 2007; Sneh and Weinberger, 2014 |
| Hula basin | Schattner and Weinberger, 2008 |
| Lebanon and Syria | Weinberger et al., 2009; Garfunkel, 2014; Sneh and Weinberger, 2014 |

*Table A3: References for faults located beyond Israel borders and/or subsurface faults*

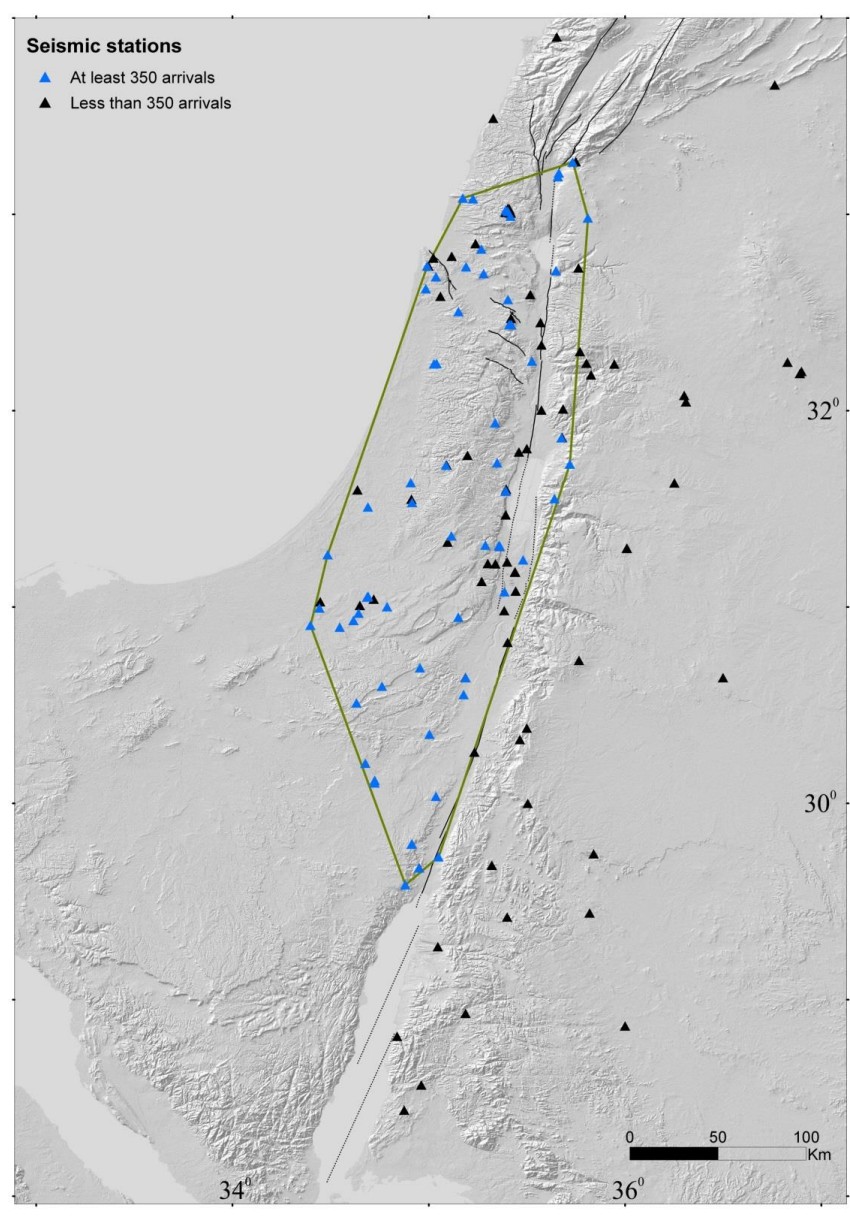

**Figure A2. Seismic stations utilised for recording the earthquakes of the examined catalogue, and the ensuing seismic network coverage area. The spatial distribution of the stations is temporal dependent. Stations that recorded less than 350 arrivals are in black, while stations that recorded more than 350 arrivals are in blue. Green lines mark the borders of the seismic network coverage area as defined in this study.**



**Figure A3. Quaternary faults superimposed on the seismicity polygons of the seismicity-based criterion.**







## 10. Appendix references

Avni, Y.: Paleogeography and tectonics of the Central Negev and the Dead Sea Rift western margin during the late Neogene and Quaternary, Ph.D. thesis, Hebrew University of Jerusalem, Geological Survey of Israel Report No. GSI/24/98, Jerusalem, 231 pp. (in Hebrew, English abstract), 1998.

Avni, Y., and Zilberman, E.; Landscape evolution triggered by neotectonics in the Sede Zin region, central Negev, Israel, Israel J. Earth. Sci., 55, 189–208, 2007.

Bartov, Y., Agnon, A., Enzel, Y., and Stein, M.: Late Quaternary faulting and subsidence in the central Dead Sea basin, Israel, Israel J. Earth Sci., 55, 17–32, 2006.

Ben-Avraham, Z.: Structural framework of the Gulf of Elat (Aqaba), Northern Red Sea, J. Geophys. Res., 90(B1), 703–726, 1985.

Ben-Avraham, Z., and Schubert, G.: Deep "drop down" basin in the southern Dead Sea, Earth Planet. Sc. Lett., 251, 254–263, 2006.

Buchbinder, B., and Sneh, A.: Marine sandstones and terrestrial conglomerates and mudstones of Neogene – Pleistocene age in the Modi'im area: a re-evaluation, Geological Survey of Israel Current Research, 1983–84, 65–69. 1984.

Calvo, R.: Stratigraphy and petrology of the Hazeva Formation in the Arava and the Negev: Implications for the development of sedimentary basins and the morphotectonics of the Dead Sea Rift Valley, Ph.D. thesis, Hebrew University of Jerusalem, Geological Survey of Israel Report No. GSI/22/02, Jerusalem, 264 pp. (in Hebrew, English abstract), 2002.

Calvo, R., Bartov, Y., Avni, Y., Garfunkel, Z., and Frislander, U.: Geological field trip to the Karkom graben: The Hazeva Fm. and its relation to the structure, Annual Meeting Field Trips Guidebook, Israel Geological Society, pp. 47–62 (in Hebrew), 1998.

Enzel, Y., Saliv, G., and Kaplan, M.: The tectonic deformation along the Zin Lineament, Nuclear Power Plant - Shivta Site: preliminary safety analysis Report. Appendix




2.5E: Late Cenozoic Geology in the Site area. Israel Electric Corporation Ltd.,
1988.

Enzel, Y., Kadan, G., and Eyal, Y.: Holocene earthquakes inferred from a Fan-Delta
sequence in The Dead Sea Graben, Quaternary Res., 53, 34–48, 2000.

Eppelbaum, L., Ben-Avraham, Z., and Katz, Y.: Structure of the Sea of Galilee and
Kinarot Valley derived from combined geological-geophysical analysis, First
Break, 25(1), 21–28, 2007.

Eyal, Y., Kaufman, A., and Bar-Matthews, M.: Use of $^{230}$Th/U ages of striated Carnotites
for dating fault displacements. Geology, 20, 829–832, 1992.

Farr, T. G., et al.: The Shuttle Radar Topography Mission, Rev. Geophys., 45, RG2004,
https://doi:10.1029/2005RG000183, 2007.

Ferry, M., Meghraoui, M., Abou Karaki, N., Al-Taj, M., Amoush, H., Al-Dhaisat, S., and
Barjous, M.: A 48-kyr-long slip rate history for the Jordan Valley segment of the
Dead Sea Fault, Earth Planet. Sc. Lett., 260, 394–406, 2007.

Garfunkel, Z.: Lateral motion and deformation along the Dead Sea transform, in: Dead
Sea Transform Fault System: Reviews, edited by: Garfunkel, Z., Ben-Avraham, Z.,
and Kagan, E. J., Springer, Dordrecht, the Netherlands, 109–150, 2014.

Ginat, H.: Paleogeography and the landscape evolution of the Nahal Hiyyon and Nahal
Zihor basins, Ph.D. thesis, Hebrew University of Jerusalem, Geological Survey of
Israel Report No. GSI/19/97, Jerusalem, 206 pp. (in Hebrew, English abstract),
1997.

Hartman, G., Niemi, T. M., Tibor, G., Ben-Avraham, Z., Al-Zoubi, A., Makovsky, Y.,
Akawwi, E., Abueladas, A.-R., and Al-Ruzouq, R.: Quaternary tectonic evolution
of the Northern Gulf of Elat/Aqaba along the Dead Sea Transform, J. Geophys.
Res.: Solid Earth, 119, 9183–9205, doi:10.1002/2013JB010879, 2014.

Hurwitz, S., Garfunkel, Z., Ben-Gai, Y., Reznikov, M., Rotstein, Y., and Gvirtzman, H.:
The tectonic framework of a complex pull-apart basin: seismic reflection





observations in the Sea of Galilee, Dead Sea transform. Tectonophysics, 359(3–4),
289–306, 2002.

IEC and WLA (Israel Electric Corporation and William Lettis & Associates, Inc.):
Shivta-Rogem Site Report. Israel Electric Corporation, Ltd., 2002.

Kafri, U., and Ecker, A.: Neogene and Quaternary subsurface geology and hydrogeology
of the Zevulun plain, Geological Survey of Israel Bulletin No. 37, Jerusalem, 13
pp., 1964.

Katz, O., Amit, R., Yagoda-Biran, G., Hatzor, Y. H., Porat, N., and Medvedev, B.:
Quaternary earthquakes and landslides in the Sea of Galilee area, the Dead Sea
Transform: paleoseismic analysis and implication to the current hazard, Israel J.
Earth. Sci., 58, 275–294, 2009.

Le Béon, M., Klinger, Y., Mériaux, A.-S., Al-Qaryouti, M., Finkel, R. C., Mayyas, O.,
and Tapponnier, P.: Quaternary morphotectonic mapping of the Wadi Araba and
implications for the tectonic activity of the southern Dead Sea fault. Tectonics, 31,
TC5003, doi:10.1029/2012TC003112, 2012.

Marco, S., Hartal, M., Hazan, N., Lev, L. and Stein, M.: Archaeology, history and
Geology of the A.D. 749 earthquake, Dead Sea transform, Geology, 31, 665– 668,
2003.

Mitchell, S. G., Matmon, A., Bierman, P. R., Enzel, Y., Caffee, M., and Rizzo, D.:
Displacement history of a limestone normal fault scarp, northern Israel, from
cosmogenic $^{36}$Cl, J. Geophys. Res., 106(B3), 4247–4264, 2001.

Reznikov, M., Ben-Avraham, Z., Garfunkel, Z., Gvirtzman, H., and Rotstein, Y.:
Structural and stratigraphic framework of Lake Kinneret, Israel J. Earth. Sci., 53,
131–149, 2004.

Sagy, A., and Nahmias, Y.: Characterizing active faulting zone, in: Infrastructure
instability along the Dead Sea: Final Report: 2008–2010, edited by: Baer, G.,
Geological Survey of Israel Report No. GSI/02/2011, Jerusalem, 7–17 (in Hebrew),
2011.





Sagy, Y., and Gvirtzman, Z.: Subsurface mapping of the Zevulun valley, The
Geophysical Institute of Israel Report 648/454/09, Lod, 21 pp. (in Hebrew), 2009.

Schattner, U., and Ben-Avraham, Z.: Transform margin of the northern Levant, eastern
Mediterranean: From formation to reactivation, Tectonics, 26, TC5020,
doi:10.1029/2007TC002112, 2007.

Schattner, U., and Weinberger, R.: A mid-Pleistocene deformation transition in the Hula
basin, northern Israel: Implications for the tectonic evolution of the Dead Sea Fault,
Geochem. Geophys. Geosyst., 9(7), Q07009, doi: 10.1029/2007GC001937, 2008.

Sneh, A., and Weinberger, R.: Major geological structures of Israel and Environs,
Geological Survey of Israel, Jerusalem, 2014.

Weinberger, R., Gross, M. R., and Sneh, A.: Evolving deformation along a transform
plate boundary: Example from the Dead Sea Fault in northern Israel, Tectonics, 28,
TC5005, doi:10.1029/2008TC002316, 2009.

Wieler, N., Avni, A., Ginat, H., and Rosensaft, M.: Quaternary map of the Eilat region on
a scale of 10:000 with explanatory notes, Geological Survey of Israel Report No.
GSI/37/2016, Jerusalem, 15 pp. (in Hebrew, English abstract), 2017.

Zilberman, E.: The geology of the central Sinai-Negev shear zone, central Negev. Part C:
The Paran Lineament, Geological Survey of Israel Report No. GSI/38/85,
Jerusalem, 53 pp., 1985.

Zilberman, E., Baer. G., Avni, Y., and Feigin, D.: Pliocene fluvial systems and tectonics
in the central Negev, southern Israel, Israel J. Earth. Sci., 45, 113–126, 1996.

Zilberman, E., Greenbaum, N., Nahmias, Y., Porat, N., and Ashqar, L.: Middle
Pleistocene to Holocene tectonic activity along the Carmel Fault - preliminary
results of a paleoseismic study, Geological Survey of Israel Report No.
GSI/02/2007, Jerusalem, 35 pp., 2006.

Zilberman, E., Greenbaum, N., Nahmias, Y., Porat, N., and Ashkar, L.: Late Pleistocene
to Holocene tectonic activity along the Nesher fault, Mount Carmel, Israel, Israel J.
Earth. Sci., 57, 87–100, 2008.



Zilberman, E., Nahmias, Y., Gvirtzman, Z., and Porat, N.: Evidence for late Pleistocene
and Holocene tectonic activity along the Bet Qeshet fault system in the Lower
Galilee, Geological Survey of Israel Report No. GSI/06/2009, Jerusalem, 22 pp. (in
Hebrew, English abstract), 2009.

Zilberman, E., Ron, H., Sa'ar, R.: Evaluating the potential seismic hazards of the Ahihud
Ridge fault system by paleomagnetic and morphological analyses of calcretes,
Geological Survey of Israel Report No. GSI/15/2011, Jerusalem, 30 pp., 2011.

