# Peer review of "Assessment of seismic sources and capable faults through"

_Natural Hazards and Earth System Sciences, 2019_

## Referee Comment (RC1) · Anonymous Referee #1 · 3 Jun 2019

The paper presents two very different types of analyses – the 'seismicity criteria', based on the earthquake kernel density and the seismic moment density, both using a 35-year time-window. The other one is mapping of active or 'capable' faults, based on geological maps and literature review. While the analyses themselves seem professional and accurate, the overall context and connection is either unclear or even misleading in places. For example – the title of the manuscript: "Assessment of potential seismic hazard for sensitive facilities" is misleading and erroneous. The paper does not contain any hazard analysis, or a comparison to existing hazard assessments for the area. There is no discussion on how these results will affect hazard or any direct practical connection between the presented analysis and hazard calculations. Moreover –

throughout the paper, surface rupture and ground shaking are intermixed as 'seismic hazard' and the fault mapping is presented as the answer for both. However – ground shaking and surface rupture are two very different types of hazard. They require different considerations in planning, etc. Is it wise to treat both as one? Seems to me that your mapping methodology is more appropriate for surface rupture analysis than for shaking (which also takes into account faults that did not rupture the surface, etc.). Please be more accurate in describing your contribution and its expected useage.

What is very much missing is a thorough discussion on the relationship between the two types of analysis (seismicity based criteria and faulting) – how do you suggest combining the two datasets that you have created ?

(1) In places where they overlap (e.g. DST), should they both be accounted for in the hazard analysis? If not – what should be the interaction ? (2) In places where they do not overlap (e.g. east Sinai), do you ignore the seismicity criterion? Do you add a 'seismogenic zone'? What is your suggestion? (3) What about places in which the kernel density is zero? Do you think there is really a zero probability of an earthquake occurring there, keeping in mind the short time window used for the kernel density? These are all very important hazard decisions, which this paper does not address.

The abstract says: "our analysis allows revealing the tectonic evolution of a given region". Therefore, it is expected that you will show this later in the results. Nowhere in the paper do you "reveal" anything new about the tectonic evolution that wasn't already known. Therefore – please clarify what exactly is new knowledge gained by this paper? This is typically done by comparing to previous studies or discussing the specific contribution presented in this study.

Other comments:

Table 1: title of 2nd column should be 'slip rate' rather than 'strike-slip'. Also, seems to me that the first slip rate that is mentioned for the Yammuneh fault is too low. It references Gomez 2007 but I think his numbers are higher. How exactly did you reach

2.8 mm/yr?

Conclusion number 3 is not exactly a conclusion. It's an opinion, or a suggestion. While important and relevant, it isn't based on any analysis or data and hence cannot be presented as a conclusion of the paper. Please rephrase.

Line 296 – the symbol Vs is typically used for shear-wave velocity in the geotechnical earthquake engineering community. I suggest using something else for slip rate.

Line 454 – remove 'many'

Line 455 – 'could have entered the map' rather than enter

Line 460 – 'Quaternary activity exists'.

Line 462 – siting of what? What is siting? Why is this related?

Please also note the supplement to this comment:
https://www.nat-hazards-earth-syst-sci-discuss.net/nhess-2019-67/nhess-2019-67-RC1-supplement.pdf

---

## Referee Comment (RC2) · Anonymous Referee #2 · 17 Jun 2019

General comments

The manuscript presents an interesting review of the existing data for Israel in terms of instrumental seismic catalog and Quaternary faults. From this database, the Authors try to derive some criteria for seismic hazard assessment. My main comment is that the manuscript objectives as described in the title and abstract are not in line with the presented data and methodology.

The database compiled by the Authors is a high quality one, still far from being complete. I find nothing new in terms of methodology. If the goal of this research is to describe a new approach for seismic hazard assessment for critical facilities, this goal

is not achieved. First of all, the manuscript deals with Israel region, and I do not see how this approach can be exported to other seismotectonic settings. Even for the Israel region, the manuscript does not address the most critical issue , that is the potential for M>6 earthquakes and accompanying surface faulting in the areas that are not close to the Dead Sea Transform, such as the Sinai peninsula and the coastal region along the Mediterranean Sea. This topic should be discussed based on the data presented in the manuscript. Several critical facilities in the Levant region are located, or are in the process of being located, relatively far from the DST. The reason is obvious, the seismic hazard along the DST is clearly very high, and whenever this is possible sites along the DST are immediately discarded during any process of siting for high risk plants and infrastructures. The manuscript should be revised in other to take into account ground motion and ground rupture hazard evaluation in the less active areas.

Moreover, the criteria used for interpreting Quaternary faults as capable faults are not very clear. Some marginal fault of the DST is interpreted as "source of M>6 earthquakes", some other as "capable fault". This is misleading. If the definition of capable fault is "a fault with significant potential for earthquake surface faulting", a fault capable of surface faulting is by definition a source for M>6 earthquakes; of course, assuming that hypocentral depth is shallow crustal, as clearly stated in Wells and Coppersmith (1994). If the problem is the probability of surface faulting events, there should be a discrimination in terms of seismotectonic setting. Along the DST, that is a very active structure, the time window to be considered for capable faults should be relatively short, like the Holocene, or 13 kyr BP (the Lisan Lake shoreline criterion used in the regulatory framework for Israel). For the Sinai region, the Quaternary criterion is much more reasonable. The choice of different time-windows for fault capability takes into account the regional plate tectonic setting of the Levant. Using the Quaternary time windows for the whole region does not.

In fact, from the historical seismicity perspective, all along the Dead Sea Transform you have a sequence of large events with epicentral intensity X or XI in the MM scale.

This implies that virtually every fault along the DST might have been reactivated by coseismic surface faulting in the past 2000 years or so. This macroseismic evidence should be properly taken into account.

Specific comments

In Figure 7 from the manuscript, there is no Quaternary fault East of the DST, and very little West of the DST. Is this a real feature, or is controlled by the completeness and resolution of the instrumental and geological database?

The manuscript describe and discuss the available instrumental earthquake catalogue. No discussion and description is available about the historical catalogue. Integration between instrumental and pre-instrumental datasets is fundamental for seismic hazard assessment. Please discuss.

Several other detailed specific comments are available in the annotated manuscript sent to the Authors.

Technical comments

A few papers cited in the text are not included in the list of references; find this in the attached annotated manuscript.

Please also note the supplement to this comment:
https://www.nat-hazards-earth-syst-sci-discuss.net/nhess-2019-67/nhess-2019-67-RC2-supplement.pdf

**Supplement:**

[revised manuscript text omitted]

---

## Referee Comment (RC3) · Anonymous Referee #3 · 18 Jun 2019

The present review contains observations and comments on "assessment of Potential seismic hazard for sensitive facilities by applying seismo-tectonic criteria: an example from the Levant region". The paper used the seismic characteristics as well as the geology and slip rates of the faults in the Levant area in order to help assessing seismic hazard. Although the paper adopts a terrific state-of-the-practice, many points have to be addressed to get an acceptable outcomes as the paper in its current state is open to criticism. 1- The title is irrelevant as I could not see a comprehensive seismic hazard analysis, unless the author consider defining the active and capable faults (seismic source model) is a complete hazard assessment process. It is very important component in any seismic hazard study, but it is not the entire process. 2- Regarding

the exclusion of a very important event Md 5.8 1993 due to unreliable location, please show how much the location is uncertain. Compare this with the clear uncertainty in location around the Gulf of Aqaba. 3- Please show how many circles are included in your calculations and the weight of each circle and why did you select (calculate) such weight. 4- It seems that the catalogue contains the aftershocks of large earthquakes, please show the role of these aftershocks as they are not due to primary tectonic movements. What is the situation if these aftershocks are removed from the calculations of earthquake kernel density distribution? 5- For many faults the slip rate is provided based upon geologic or GPS surveys. Such slip may contain a creep component in addition to the seismic one. The role of creep should be addressed for all active faults as it could be a source of large uncertainty. 5- With the large periods of quiescence observed frequently along many parts of DST, 35 years of instrumentally recorded seismicity are very short to reflect the active tectonics accurately. This period should be extended using robust historical records. 6- Although the seismicity and earthquake kernel density distribution show high seismicity to the east of the Gulf of Aqaba, neither active nor capable faults are inferred at this area. 7- Abbreviations should be explained at their first appearance in the text (e.g. LRB). Some abbreviations has no explanation (QFMI). 8- Minor comments a) Line 54 contains two fullstops, please remove one of them. b) Arrange references in line 116 in a chronological order. c) Sentence in lines 147 and 148 needs reference. d) Change figure 4 into Fig. 4 in line 258. e) Change demonstrates into demonstrate in line 282. F) Change is represents in line 366 into represents. g) Rewrite lines 408 to 411 as it is really so difficult to be followed. h) Remove many in line 454. i) Sea of Galilee should be shown on a map. j) ARF is repeated in Fig. 7. k) All the maps lack to the North Direction Indicator.

---

## Author Comment (AC1) · 7 Aug 2019

We would like to thank the unanimous reviewer for his/her in-depth review of the manuscript and his/her constructive and important comments. Following the comments, we have thoroughly revised the article. The manuscript title, introduction, discussion and conclusion chapters were rewritten. We provide below detailed replies to the reviewer's comments and indicate how and where changes were made in the revised manuscript.

1) "the title of the manuscript: "Assessment of potential seismic hazard for sensitive facilities" is misleading and erroneous. The paper does not contain any hazard analy-

sis, or a comparison to existing hazard assessments for the area" Author's response: following the reviewer comment we wrote a new title that reflect this study more accurately.

New title: "Assessment of seismic sources and capable faults through hierarchic tectonic criteria: implications for seismic hazard in the Levant "

2) "There is no discussion on how these results will affect hazard or any direct practical connection between the presented analysis and hazard calculations."

Author's response: We added a section that clarify this important subject to the discussion (See below Sec. 7.1). Indeed, we are not operating hazard statistical calculations. We hope that our new introduction clarify this. The map and the slip rates of Fig. 5, as well as the local seismic intensity that we analysed here, are fundamental inputs for ground shake models and acceleration maps. The Capable faults map, on the other hand, can be used for choosing potential cites for planning special facilities. defining faults parameters, maps and local seismic characters, as we done here, are the first steps in hazard evaluations. We further emphasize that the two maps (Figs. 5, 7) enable defining the relevant faults necessary for regional hazard models, but not necessarily replacing local maps of other faults that required in some standards, when siting in a specific location is considered.

[revised manuscript text omitted]

3) ""surface rupture and ground shaking are intermixed as 'seismic hazard' and the fault mapping is presented as the answer for both. However – ground shaking and surface rupture are two very different types of hazard. They require different considerations in planning, etc. Is it wise to treat both as one? "

Author's response: Following the reviewer comment, we declare in the introduction that we generate a database of faults that is relevant for several seismic hazards. We demonstrate how we categorize faults for two specific different requirements: one that is aimed to be used in ground shaking models, and the other for siting special infrastructures. We however do not evaluate seismic hazard in this paper, as well as site specific requirements. These are beyond the scope of this paper.

4) "What is very much missing is a thorough discussion on the relationship between the two types of analysis (seismicity based criteria and faulting) – how do you suggest combining the two datasets that you have created ?" (1) In places where they overlap

(e.g. DST), should they both be accounted for in the hazard analysis? If not – what should be the interaction ? (2) In places where they do not overlap (e.g. east Sinai), do you ignore the seismicity criterion? Do you add a 'seismogenic zone'? What is your suggestion? (3) What about places in which the kernel density is zero? Do you think there is really a zero probability of an earthquake occurring there, keeping in mind the short time window used for the kernel density? These are all very important hazard decisions, which this paper does not address."

Author's response: The products of the seismologic analysis are applied differently in the two maps (Figs. 5, 7). We design a seismicity-based criterion that is based on the distribution of two parameters: the earthquake kernel density and the seismic moment kernel density. Faults which are located beyond this pattern are not part of the faults of Fig. 5. The seismological character of a zone is considered as part of criterion for the map in Fig. 7. The success of this selection is further reinforced by the match between the geological-categorized faults and the seismicity criterion (Fig. A3). If this comment, and the three options listed by the referee, refer to the aspect of utilizing the 'gridded seismicity', as an independent database for both surface rupture and ground motion hazards, we emphasis that we focused on generating a database of faults and not on utilizing the seismicity as an independent source for hazard estimations. Therefore, we did not discuss this issue. We now clarify this in the introduction. However, we add a section that discuss the applications of our different analyses to seismic hazard evaluation, included possible usages of the 'gridded seismicity' for hazard evaluations (lines 474-480).

5) "Seems to me that your mapping methodology is more appropriate for surface rupture analysis than for shaking (which also takes into account faults that did not rupture the surface, etc.). Please be more accurate in describing your contribution and its expected useage."

Author's response: A discussion focus on our methodology and its efficiency to both different aims is now written in Sec. 7.1 (See above). Subsurface faults were considered

for capable faults if they are continuation of categorised faults or if there is information that they offset Quaternary formations. On the other hand, for the main seismic sources, they are neglected. Indeed when a local siting process is applied (both for rupture surface and for shaking), information on local fault which are not categorised in our regional analysis should be taking into account.

6) " The abstract says: "our analysis allows revealing the tectonic evolution of a given region". Therefore, it is expected that you will show this later in the results. Nowhere in the paper do you "reveal" anything new about the tectonic evolution that wasn't already known. Therefore – please clarify what exactly is new knowledge gained by this paper? This is typically done by comparing to previous studies or discussing the specific contribution presented in this study.

Author's response: We changed this sentence in the abstract. We also added a new section (Sec. 7.2 in our new discussion), A new figure in the appendix (A4), and conclusion (N. 6, See below) focuses on implications for local tectonics and slip dynamics.

7) "Table 1: title of 2nd column should be 'slip rate' rather than 'strike-slip'. Also, seems to me that the first slip rate that is mentioned for the Yammuneh fault is too low. It references Gomez 2007 but I think his numbers are higher. How exactly did you reach 2.8 mm/yr?"

Author's response: Numbers are for lateral slip rates. 2.8 mm/yr in Table 1 was a mistake and deleted.

8) "Conclusion number 3 is not exactly a conclusion. It's an opinion, or a suggestion. While important and relevant, it isn't based on any analysis or data and hence cannot be presented as a conclusion of the paper. Please rephrase."

Author's response: Following the three reviewers comments we rewrite the entire conclusions section.

"8. Conclusions 1. Mapping and characterizing faults that pose seismic hazard, particularly in regions with intermediate seismicity rates and/or where young formations are sparse, require developing an interdisciplinary regional database and hierarchical seismo-tectonic criteria. With respect to the specific dictated requirements, faults that are potential sources for the far-field and for the near-field (i.e., surface rupture) hazards should be analyzed by different criteria; both represent seismic hazard of significant earthquakes, but within different time frames. 2. We design a seismicity-based criterion that using the distribution of two parameters: the earthquake kernel density and the seismic moment kernel density. The success of this selection is demonstrated by the match between the geological-categorized faults and the seismicity criterion (Fig. A3). The union zone defined by these two statistical distributions is efficient in both definition of the main seismic sources (Fig. A3) and in categorizing capable faults (Fig. 7). 3. The hierarchic seismo-tectonic criteria ideally reflects the degree of certainty for recent faulting, and can later be implemented if a specific hazard is considered or if risk evaluation is applied. 4. The temporal reference for local planning of critical facilities such as dams and nuclear power plants is usually long, because the possible damage to the construction has severe regional implications. We selected the Quaternary period as the relevant time frame for capable faults in the region of Israel. While this time frame (2.6 Ma) is longer than the previous for defining capable faults for a potential local nuclear power plant (IEC and WLA, 2002), it is justified by considering the regional stress field, the regional stratigraphic configurations and the criteria that focus on surface rupture rather than general fault movements. We suggest that tectonic and stratigraphic conditions, as well as the accessibility of geologic maps and their resolutions, should be considered for defining the time frame for capable faults. 5. Beyond planning of special constructions, the developed database and the maps that are generated and presented here constitute further applications for planning and research. The regional main seismic sources map (Fig. 5) is fundamental for seismotectonic modelling and eventually for generating ground motion prediction maps (e.g. by PSHA) that include essential information for construction planning. The capable fault database and the related maps (Figs. 2-4, 6-7) lay the foundation for further

study of the regional Quaternary faulting and tectonics in the Israel region. Further-more, the methodology, which is based on categorization and sub-categorization by seismo-tectonic hierarchic criteria, enables differentiation of hazard potential and can be applied in other regions around the world. 6. Compering instrumental seismicity with geodetic slip rates enables to reveal seismo-tectonic patterns in an investigated region. Specifically, we recognize along the DST zones of enhanced or reduced seis-micity, which can be controlled either by slip partitioning, creep or litho-structural com-plexities in fault junctions. In addition, ∼NW orientated seismic activity was identified branched from the DST (EBL in Fig. 6). This activity might reflects reactivation of extensional feature developed during the post-Eocene Red Sea rifting. "

9) "Line 296 – the symbol Vs is typically used for shear-wave velocity in the geotechni-cal earthquake engineering community. I suggest using something else for slip rate."

Author's response: We no longer use this symbol. Instead, we now use a simple "range" (e.g. a – b mm/yr) Author's changes in manuscript:

"these faults are associated with Holocene average sinistral slip rates of 1 – 5 mm/yr."

10) " Line 454 – remove 'many' "

Answer: Removed 11) " Line 455 – 'could have entered the map' rather than enter "

Answer: Corrected

12) "Line 460 – 'Quaternary activity exists'."

Answer: Corrected

13) "Line 462 – siting of what? What is siting? Why is this related?"

Author's response: We rewrite the entire section.

---

## Author Comment (AC2) · 7 Aug 2019

We would like to thank the unanimous reviewer for his/her in-depth review of the manuscript and his/her constructive and important comments. Following the comments, we have thoroughly revised the article. The manuscript title, introduction, discussion and conclusion chapters were rewritten. We provide below detailed replies to the reviewer's comments and indicate how and where changes were made in the revised manuscript.

1) "I find nothing new in terms of methodology."

Author's response: The reviewer is right that seismo-tectonic criteria for categorizing faults were previously applied during seismic hazard analyses. However, in addition to classifying hazardous faults by the recency of faulting or their recurrence intervals: a) We design a seismicity-based criterion that use the distribution of two parameters: the Earthquake Kernel Density and the Seismic Moment Kernel Density. The success of this selection is further reinforced by the match between the geological-categorized faults and the seismicity criterion (Fig. A3). b) Seismic sources for ground shaking maps are considered only faults that are satisfied both geological and seismological criterions. This is significant when slip rates are mostly unknown (as in Sec. 5.2). c) The internal hierarchic categorization of faults, in both maps enable weighting different faults when hazard analysis is applied.

We added a section summarize the above points to the discussion (Sec. 7.1, Please see our response to reviewer 1).

2) "If the goal of this research is to describe a new approach for seismic hazard assessment for critical facilities, this goal is not achieved"

Author's response: We agree and following the comment of the Anonymous Referee #1, we have already changed the title of this paper, so it is no longer "assessment of potential seismic hazard".

3) "First of all, the manuscript deals with Israel region, and I do not see how this approach can be exported to other seismo-tectonic settings."

Author's response: We now specifically discuss the universal aspects of our analysis in Sec. 7.1 (Please see our response to reviewer 1.

4) "Even for the Israel region, the manuscript does not address the most critical issue , that is the potential for M>6 earthquakes and accompanying surface faulting in the areas that are not close to the Dead Sea Transform, such as the Sinai peninsula and the coastal region along the Mediterranean Sea. This topic should be discussed based on the data presented in the manuscript. Several critical facilities in the Levant region are located, or are in the process of being located, relatively far from the DST. The reason is obvious, the seismic hazard along the DST is clearly very high, and whenever this is possible sites along the DST are immediately discarded during any process of siting for high risk plants and infrastructures. The manuscript should be revised in other to take into account ground motion and ground rupture hazard evaluation in the less active areas."

Author's response: The potential of capable faults, which are not part of the main seismic sources, is indeed not discuss in this paper. The aim of this paper was to separate the capable faults from other faults and categorize them. The next step should be generating statistic and/or deterministic methods for defining the safety distances for any specific siting. This is beyond the scope of the paper.

5) "Moreover, the criteria used for interpreting Quaternary faults as capable faults are not very clear. Some marginal fault of the DST is interpreted as "source of M>6 earthquakes", some other as "capable fault". This is misleading. If the definition of capable fault is "a fault with significant potential for earthquake surface faulting", a fault capable of surface faulting is by definition a source for M>6 earthquakes; of course, assuming that hypocentral depth is shallow crustal, as clearly stated in Wells and Coppersmith (1994).

Author's response: The faults in the different maps (Figs. 5, 7) are all capable to generate large earthquake (M>6), however the time frame, slip rates and seismological activity are different. Specific definitions are presented and explained in the beginning of Sec. 5 and Sec. 6.1 "Framework and principles".

6) "If the problem is the probability of surface faulting events, there should be a discrimination in terms of seismo-tectonic setting. Along the DST, that is a very active structure, the time window to be considered for capable faults should be relatively short, like the Holocene, or 13 kyr BP (the Lisan Lake shoreline criterion used in the regulatory framework for Israel). For the Sinai region, the Quaternary criterion is much more reasonable. The choice of different time-windows for fault capability takes into account the regional plate tectonic setting of the Levant. Using the Quaternary time windows for the whole region does not."

Author's response: Basically, we think that using longer time intervals for defining capable faults as the distance from active sources increase can be misleading. Even using the "earthquake cycle" period, for defining capable faults, as suggested by Machette (2000), might be sustained only when large regimes are compared. We (see Sec. 6.1) suggests that the combination of the tectonic regional field (stress field orientation, displacement rates) and the level of the geological information (stratigraphy, map resolutions) should determine the relevant time frame for capable faults.

7) "In fact, from the historical seismicity perspective, all along the Dead Sea Transform you have a sequence of large events with epicentral intensity X or XI in the MM scale. This implies that virtually every fault along the DST might have been reactivated by coseismic surface faulting in the past 2000 years or so. This macroseismic evidence should be properly taken into account."

Author's response: The estimated Intensity of past earthquakes is translated into magnitudes. We already considered the interpretation of historical earthquakes for estimating the maximum magnitude at the end of Sec. 5.1: "Previous analyses of maximum earthquake magnitude based on historical earthquakes or on background seismicity predicted magnitudes of $\leq 7.8$ Mw for the largest segments (e.g., Stevens and Avouac., 2017; Klinger et al., 2015; Hamiel et al., 2018a)."

We think that these references and the related slip rates are sufficient for this paper purpose.

8) "In Figure 7 from the manuscript, there is no Quaternary fault East of the DST, and very little West of the DST. Is this a real feature, or is controlled by the completeness and resolution of the instrumental and geological database?"

Author's response: Indeed the Capable fault map not include faults in neighboring countries. We now clearly declare in Sec. 6.1 "Finally, because of the limitation of our database, mapped capable faults (Fig. 7) are limited to Israel region, unless their continuations spread to the neighboring countries." ; also see our geological database in Sec. 3.

9) "The manuscript describe and discuss the available instrumental earthquake catalog. No discussion and description is available about the historical catalog. Integration between instrumental and pre-instrumental datasets is fundamental for seismic hazard assessment. Please discuss."

Author's response: We regard the information of historical earthquakes for estimating the maximum magnitude. Further considerations are beyond the scope of this paper. We now changed the title of our manuscript so it is no longer "seismic hazard assessment".

10) "A few papers cited in the text are not included in the list of references; find this in the attached annotated manuscript."

Author's response: Fixed

11) "Please also note the supplement to this comment: https://www.nat-hazards-earth-syst-sci-discuss.net/nhess-2019-67/nhess-2019-67-RC2-supplement.pdf"

Author's response: We responded to the comments in this pdf file, and clarify associated issues in our new version of the manuscript.

Please also note the supplement to this comment:
https://www.nat-hazards-earth-syst-sci-discuss.net/nhess-2019-67/nhess-2019-67-AC2-supplement.pdf

**Supplement:**

[revised manuscript text omitted]

---

## Author Comment (AC3) · 7 Aug 2019

We would like to thank the unanimous reviewer for his/her in-depth review of the manuscript and his/her constructive and important comments. Following the comments, we have thoroughly revised the article. The manuscript title, introduction, discussion and conclusion chapters were rewritten. We provide below detailed replies to the reviewer's comments and indicate how and where changes were made in the revised manuscript.

"1- The title is irrelevant as I could not see a comprehensive seismic hazard analysis, unless the author consider defining the active and capable faults (seismic source

model) is a complete hazard assessment process. It is very important component in any seismic hazard study, but it is not the entire process."

Author's response: We accept this comment. We have changed the title of this paper. New title: "Assessment of seismic sources and capable faults through hierarchic tectonic criteria: implications for seismic hazard in the Levant"

"2- Regarding the exclusion of a very important event Md 5.8 1993 due to unreliable location, please show how much the location is uncertain. Compare this with the clear uncertainty in location around the Gulf of Aqaba."

Author's response: The event Md 5.8 1993 is indeed important, but we note that it occurred in the most seismically active zone of the Gulf of Elat (Aqaba), in such distance from the inland mapped faults, that it would not have any impact on the current results of the mapped fault. Since the focus in this paper is the mapping of faults, we think that adding further details such as error issues of a specific earthquake, which has no impact on the mapping of any fault, is irrelevant. We now clarify this though.

Author's changes in manuscript (lines 189-190): "large location errors (including the $M\_d=5.8$ 1993 event in the Gulf of Elat, which anyhow does not affect the marking of faults in this paper since it was nucleated outside our high-resolution geological data)."

"3- Please show how many circles are included in your calculations and the weight of each circle"

Author's response: There 'circles' were given as an illustration, as mentioned in the text. However, we now see this illustration might be confusing, so we rephrase:

Author's changes in manuscript: "This circular-shape based approach prevents any directional bias. "

the continuation of the last comment: "and why did you select (calculate) such weight."

Author's response: see lines 213-218:

"The 6-km limitation, the Gaussian function and its standard deviation of 2 (for the kernel estimation), were tuned and chosen to: a) capture different seismic patches along active faults; b) be significantly larger than the location horizontal median error (∼1.2 km; Wetzler and Kurzon, 2016); c) assign higher weight to events closer to the evaluated grid-point; d) include as many events as possible for achieving statistical significance at each of the grid-points. " "4- It seems that the catalog contains the aftershocks of large earthquakes, please show the role of these aftershocks as they are not due to primary tectonic movements. What is the situation if these aftershocks are removed from the calculations of earthquake kernel density distribution?"

Author's response: Indeed, the catalog contains the aftershocks of medium to large earthquake. As already mentioned, the focus of this paper is the mapping of faults, based on the suggested methodology of hierarchic seismo-tectonic criteria. Showing the exact role of the aftershocks, and the situation if they are removed from the calculation, is a different topic. We also note that aftershocks may be also associated with reactivation or even surface rupture, so they should not be neglected in a seismicity-based criterion for a capable fault map.

"5- For many faults the slip rate is provided based upon geologic or GPS surveys. Such slip may contain a creep component in addition to the seismic one. The role of creep should be addressed for all active faults as it could be a source of large uncertainty."

Author's response: We had already addressed the issue of evidences for a creep component: see discussion (lines 516-523) and Table 1 (lines 790, 794):

"5- With the large periods of quiescence observed frequently along many parts of DST, 35 years of instrumentally recorded seismicity are very short to reflect the active tectonics accurately. This period should be extended using robust historical records."

Author's response: We regard the information of historical earthquakes for estimating the maximum magnitude (lines 316-318):

"6- Although the seismicity and earthquake kernel density distribution show high seismicity to the east of the Gulf of Aqaba, neither active nor capable faults are inferred at this area."

Author's response: As we already noted (see response to reviewer 2), the capable fault map, except a few exceptions (see Sec. 3; 6) does not include the the neighboring countries. Specifically, we mapped a few faults within the Gulf of Aqaba that we define as 'main seismic sources' (see Sec. 5). This is sufficient for our purposes, which we now describe more clearly through the paper.

"7- Abbreviations should be explained at their first appearance in the text (e.g. LRB). Some abbreviations has no explanation (QFMI)."

Author's response: Corrected.

"8- Minor comments a) Line 54 contains two fullstops, please remove one of them. b) Arrange references in line 116 in a chronological order. c) Sentence in lines 147 and 148 needs reference. d) Change figure 4 into Fig. 4 in line 258. e) Change demonstrates into demonstrate in line 282. F) Change is represents in line 366 into represents. g) Rewrite lines 408 to 411 as it is really so difficult to be followed. h) Remove many in line 454. i) Sea of Galilee should be shown on a map. j) ARF is repeated in Fig. 7. k) All the maps lack to the North Direction Indicator."

Author's response:

Two full-stops are now removed in all parts of the manuscript. Corrected (now in line 119) Author's changes in manuscript: "(e.g. Amit et al., 2002; Marco et al., 2005; Marco, 2008)." We now add a new reference (now in line 185): Author's changes in manuscript: "Moreover, the boundary Pleistocene-Pliocene (Neogene-Quaternary) was shifted in 2009, from ∼1.8Ma to ∼2.6Ma (Gibbard et al., 2010)". Corrected (now in line 267): Author's changes in manuscript: "[...] is shown in Fig. 4" Corrected (now in line 274): Author's changes in manuscript: "faults that demonstrate slip rates [...]" Corrected ("is" is deleted, now in line 375): Author's changes in manuscript: "The regional stress state within the Quaternary period represents well the current stress field" We rephrased (now in lines 416-419): Author's changes in manuscript: "The polygons are defined by a threshold value, so that each of them is the smallest to cover continuously the whole length of the most active tectonic feature in the region. In this case study, this feature is the DST, but we exclude the relatively silent northern section of the Jordan Valley segment" Already removed due to comment by Anonymous Reviewer #1 We added marks for the Dead Sea Basin (DSB) and for the Sea of Galilee (SG) to figure one. Fixed. All our maps are orientated such that the north is directed upwards. We do not think that north arrow is necessary.
* * *

---

## Author Response (AR1)

| 1 | [paper: nhess-2019-67] |
|---|------------------------|
|---|------------------------|

Old Title: Assessment of potential seismic hazard for sensitive facilities by

- 3 applying seismo-tectonic criteria: an example from the Levant region
- 4
- 5 New title: Assessment of seismic sources and capable faults through
- 6 hierarchic tectonic criteria: implications for seismic hazard in the Levant
- 7 Matty Sharon et al.

asagy@gsi.gov.il

We would like to thank the three anonymous reviewers for their in-depth review of the 10 manuscript and their constructive and important comments. Following the comments, we 11 have thoroughly revised the article. The manuscript title, introduction, discussion and 12 conclusion chapters were rewritten. We provide below detailed replies to the reviewer's 13 comments and indicate how and where changes were made in the revised manuscript. 14 Please note that the lines we refer to, are at the manuscript submitted as a different file. 15

17

**Referee #1**

1) "the title of the manuscript: "Assessment of potential seismic hazard for sensitive facilities" is 19 misleading and erroneous. The paper does not contain any hazard analysis, or a comparison to 20 existing hazard assessments for the area"

Author's response:

following the reviewer comment we wrote a new title that reflect this study more accurately. New title: "Assessment of seismic sources and capable faults through hierarchic tectonic criteria:
 implications for seismic hazard in the Levant "

2) "There is no discussion on how these results will affect hazard or any direct practical connection between the presented analysis and hazard calculations."

Author's response: Although our new title and introduction now describe that hazard calculations are not part of this study, we added a new section that discusses the 31 32 applications for hazard evaluations (Sec. 7.1 - 1 lines 442 - 494). The map and the slip rates of Fig. 5, as well as the local seismic intensity that we analysed here, are fundamental 33 inputs for ground shake models and acceleration maps. The capable faults map, on the 34 other hand, can be used for choosing potential cites for planning special facilities. Defining 35 faults parameters, maps and local seismic characters, as we done here, are the first steps in 36 hazard evaluations. We further emphasise that the two maps (Figs. 5, 7) enable defining 37 38 the relevant faults necessary for regional hazard models, but they do not necessarily replace local maps of other faults that required in some standards, when siting in a specific location 39

is considered. 40

3) ""surface rupture and ground shaking are intermixed as 'seismic hazard' and the fault mapping 44 is presented as the answer for both. However – ground shaking and surface rupture are two very different types of hazard. They require different considerations in planning, etc. Is it wise to treat 45 46 both as one? "

Author's response: Following the reviewer comment, we declare in the introduction (within lines 38-58) that we generate a database of faults that is relevant for several seismic 49 hazards. We demonstrate how we categorise faults for two specific different requirements: 50 51 one that is aimed to be used in ground shaking models, and the other for siting critical 52 facilities or special infrastructures. We however do not evaluate seismic hazard in this

- paper, as well as site specific requirements. These are beyond the scope of this paper. 53
- 54
- 55
- 56 57

4) "What is very much missing is a thorough discussion on the relationship between the two types of

- 58 analysis (seismicity based criteria and faulting) – how do you suggest combining the two datasets 59 that
- 60 you have created ?

(1) In places where they overlap (e.g. DST), should they both be accounted for in the hazard analysis? If not – what should be the interaction ? (2) In places where they do not overlap (e.g. 62 east Sinai), do you ignore the seismicity criterion? Do you add a 'seismogenic zone'? What is your 63 suggestion? (3) What about places in which the kernel density is zero? Do you think there is really 64

a zero probability of an earthquake occurring there, keeping in mind the short time window used for the kernel 67 density?

These are all very important hazard decisions, which this paper does not address."

Author's response: The products of the seismologic analysis are applied differently in the two maps (Figs. 5, 7). We design a seismicity-based criterion that is based on the 71 72 distribution of two parameters: the earthquake kernel density and the seismic moment 73 kernel density. Faults that are located beyond this pattern are not part of the faults of Fig. 74 5. The seismological character of a zone is considered as part of criterion for the map in 75 Fig. 7. The success of this selection is further reinforced by the match between the 76 geological-categorised faults and the seismicity criterion (Fig. A4). This subject is discussed in Sec. 7.1 (within lines 449-464). However, if this comment refers to the aspect 77 of utilising the 'gridded seismicity' (i.e. the grid-based distribution of the seismicity 78 79 parameters) as an independent database for both surface rupture and ground motion 80 hazards, we emphasise that we focus on generating databases of faults and not on utilising 81 the seismicity as an independent source for hazard evaluations. Therefore, we did not

- discuss this issue. We now clarify this in the introduction. However, we add a section that 82 83
- discuss the applications of our different analyses to seismic hazard evaluation, included
- possible usages of the 'gridded seismicity' for hazard evaluations (lines 473-476). 84
- 85

5) "Seems to me that your mapping methodology is more appropriate for surface rupture

- 87 analysis than for shaking (which also takes into account faults that did not rupture the surface,
- 88 etc.). Please be more accurate in describing your contribution and its expected useage."
- 89
- 90 Author's response: A discussion focuses on our methodology and its applications for both 91 surface rupture and ground shaking hazard analyses is now written in Sec. 7.1 (particularly relevant to this comment are lines 449-451; 465-476). Subsurface faults were considered 92 for capable faults if they are the continuation of categorised faults or if there is information 93 94 that they offset Quaternary formations. On the other hand, for the main seismic sources, 95 they are neglected. Indeed, when a local siting process is applied (both for rupture surface and for shaking), information on local fault which are not categorised in our regional 96
- 97 analysis should be taking into account.
- 98 6) "The abstract says: "our analysis allows revealing the tectonic evolution of a given region".
- 99 Therefore, it is expected that you will show this later in the results. Nowhere in the paper do
- 100 you "reveal" anything new about the tectonic evolution that wasn't already known. Therefore – 101 please clarify what exactly is new knowledge gained by this paper? This is typically done by
- 102 comparing to previous studies or discussing the specific contribution presented in this study."
- 103

Author's response: We changed this sentence in the abstract. We also added a new 105 section (Sec. 7.2 in our new discussion – lines 496 - 537), a new figure in the appendix (A4) and a conclusion (No. 6, lines 575 - 583) that focuses on the implications for local 106 tectonics and slip dynamics. 107

7) "Table 1: title of 2nd column should be 'slip rate' rather than 'strike-slip'. Also, seems to me 110 that the first slip rate that is mentioned for the Yammuneh fault is too low. It references Gomez 111 2007 but I think his numbers are higher. How exactly did you reach 2.8 mm/yr?"

- 112
- 113 Author's response: Numbers are for lateral slip rates – we now changed 'Lateral slip rate' 114 (Table 1 is in line 845). 2.8 mm/yr in Table 1 was a mistake and is now deleted.
- 115

8) "Conclusion number 3 is not exactly a conclusion. It's an opinion, or a suggestion. While 116 117 important and relevant, it isn't based on any analysis or data and hence cannot be presented as a conclusion of the paper. Please rephrase." 118

- 119
- 120 Author's response: following the three reviewers' comments we rewrote the conclusion's
- 121 section (lines 539 – 583). Specifically, we also rephrased conclusion 3 (now is listed as
- 122 conclusion  $4 - \underline{\text{lines } 556 - 565}$ ).
- 123

| 124
| 9) "Line 296 – the symbol Vs is typically used for shear-wave velocity in the geotechnical earthquake engineering community. I suggest using something else for slip rate." |
|-------------------|-----------------------------------------------------------------------------------------------------------------------------------------------------------------------------|
| 127               | Author's response: We no longer use this symbol. Instead, we now use a simple "range"                                                                                       |
| 128
| character (e.g. $a - b mm/yr$ ) (e.g. lines 302-303 ).                                                                                                               |
| 130               | 10) " Line 454 – remove 'many' "                                                                                                                                            |
| 131               |                                                                                                                                                                             |
| 132               | Author's response: Removed                                                                                                                                                  |
| 133               |                                                                                                                                                                             |
| 134
| 11) " Line 455 – 'could have entered the map' rather than enter "                                                                                                           |
| 136               | Author's response: Corrected                                                                                                                                                |
| 137               |                                                                                                                                                                             |
| 138
| 12) "Line 460 – 'Quaternary activity exists'."                                                                                                                              |
| 140
| Answer: Corrected                                                                                                                                                           |
| 142
| 13) "Line 462 – siting of what? What is siting? Why is this related?"                                                                                                       |
| 144               | Author's response: We rewrote the entire section (this subject is discussed in Sec. 7.1),                                                                                   |
| 145               | and also added relevant parts in the introduction (e.g. $lines 43 - 52$ ).                                                                                                  |
| 146               |                                                                                                                                                                             |
| 147               |                                                                                                                                                                             |
| 148               | Referee #2                                                                                                                                                                  |
| 149               |                                                                                                                                                                             |
| 150               | 1) "I find nothing new in terms of methodology."                                                                                                                            |
| 151               | Author's response: Indeed, seismo-tectonic criteria for categorising faults were                                                                                            |
| 152               | previously applied in seismic hazard analyses. However, in addition of classifying                                                                                          |
| 153               | hazardous faults by the recency of faulting or their recurrence intervals:                                                                                                  |
| 154               | a) We design a seismicity-based criterion that use the distribution of two parameters:                                                                                      |
| 155               | the earthquake kernel density and seismic moment kernel density. This criterion is                                                                                          |
| 156               | reinforced by the match between the geological-categorised faults and the seismicity                                                                                        |
| 157               | criterion (Fig. A3). b) Seismic sources for ground shaking maps are considered only                                                                                         |
| 158               | faults that are satisfied both geological and seismological criteria. This is significant                                                                                   |
| 159               | when slip rates are mostly unknown (as in Sec. 5.2).                                                                                                                        |
| 160               | c) The internal hierarchic categorisation of faults, in both maps, enable weighting                                                                                         |
| 161               | different faults when hazard evaluation is applied.                                                                                                                         |

We now discuss this in the new section 7.1.

164

2) "If the goal of this research is to describe a new approach for seismic hazard assessment166 for critical facilities, this goal is not achieved"

Author's response: We agree and following the comment of the Anonymous Referee

*#*1, we have already changed the title of this paper, so it is no longer "assessment of potential seismic hazard".

3) "First of all, the manuscript deals with Israel region, and I do not see how this approachcan be exported to other seismo-tectonic settings."

Author's response: We now specifically discuss the universal aspects of our analysis in Sec. 7.1 (also, please see our response to Referre #1).

4) "Even for the Israel region, the manuscript does not address the most critical issue, that is the potential for M>6 earthquakes and accompanying surface faulting in the areas that are not close 179 to the Dead Sea Transform, such as the Sinai peninsula and the coastal region along the 180 181 Mediterranean Sea. This topic should be discussed based on the data presented in the manuscript. Several critical facilities in the Levant region are located, or are in the process of being located, 182 183 relatively far from the DST. The reason is obvious, the seismic hazard along the DST is clearly very 184 high, and whenever this is possible sites along the DST are immediately discarded during any 185 process of siting for high risk plants and infrastructures. The manuscript should be revised in other 186 to take into account ground motion and ground rupture hazard evaluation in the less active areas." 187

Author's response: The potential of capable faults, which are not part of the main seismic sources, is indeed not discuss in this paper. The aim of this paper was to separate the capable faults from other faults and categorise them. The next step should be generating statistic and/or deterministic methods for defining the safety distances for any specific siting. This is beyond the scope of the paper.

5) "Moreover, the criteria used for interpreting Quaternary faults as capable faults are not very clear. Some marginal fault of the DST is interpreted as "source of M>6 earthquakes", some other as "capable fault". This is misleading. If the definition of capable fault is "a fault with significant potential for earthquake surface faulting", a fault capable of surface faulting is by definition a source for M>6 earthquakes; of course, assuming that hypocentral depth is shallow crustal, as clearly stated in Wells and Coppersmith (1994).

Author's response: The faults in the different maps (Figs. 5, 7) are all capable to generate large earthquake (M>6). However, the time frame, slip rates and seismological activity are different. Specific definitions are presented and explained in the beginning of Sec. 5 and Sec. 6.1 "Framework and principles".

6) "If the problem is the probability of surface faulting events, there should be a discrimination in terms of seismo-tectonic setting. Along the DST, that is a very active structure, the time window to be considered for capable faults should be relatively short, like the Holocene, or 13 kyr BP (the Lisan Lake shoreline criterion used in the regulatory framework for Israel). For the Sinai region, the Quaternary criterion is much more reasonable. The choice of different time-windows for fault capability takes into account the regional plate tectonic setting of the Levant. Using the Quaternary time windows for the whole region does not."

Author's response: Basically, we think that using longer time intervals for defining capable faults, as an increasing distance from active sources, can be misleading. Even using the "earthquake cycle" period for defining capable faults, as suggested by Machette (2000), might be sustained only when large regimes are compared. We suggest that the combination of the tectonic regional field (stress field orientation, displacement rates) and the level of the geological information (stratigraphy, map resolutions) should determine the relevant time frame for capable faults (see Sec. 6.1).

7) "In fact, from the historical seismicity perspective, all along the Dead Sea Transform you have
a sequence of large events with epicentral intensity X or XI in the MM scale. This implies that
virtually every fault along the DST might have been reactivated by coseismic surface faulting in
the past 2000 years or so. This macroseismic evidence should be properly taken into account."

Author's response: The estimated intensity of past earthquakes is translated into magnitudes. We already considered the interpretation of historical earthquakes for estimating the maximum magnitude at the end of Sec. 5.1 (lines 316 - 318). These interpretations and the related slip rates are sufficient for the purpose of this paper.

8) "In Figure 7 from the manuscript, there is no Quaternary fault East of the DST, and very little
West of the DST. Is this a real feature, or is controlled by the completeness and resolution of the
instrumental and geological database?"

Author's response: Indeed, the capable fault map does not include faults in neighboring countries. We now clearly this in Sec. 6.1 (lines 387-388). Also see our geological database in Sec. 3.

9) "The manuscript describe and discuss the available instrumental earthquake catalog. No discussion and description is available about the historical catalog. Integration between

| 242
     | instrumental and pre-instrumental datasets is fundamental for seismic hazard assessment.
Please discuss."                                                                                                                                                                                                                |
|---------------------------------|-----------------------------------------------------------------------------------------------------------------------------------------------------------------------------------------------------------------------------------------------------------------------------------------------------------------------------|
| 245                             | Author's response: We regard the information of historical earthquakes for estimating the                                                                                                                                                                                                                                   |
| 246
            | maximum magnitude. Further considerations are beyond the scope of this paper. We now changed the title of our manuscript so it is no longer "seismic hazard assessment".                                                                                                                                                    |
| 248                             |                                                                                                                                                                                                                                                                                                                             |
| 249
     | 10) "A few papers cited in the text are not included in the list of references; find this in the attached annotated manuscript."                                                                                                                                                                                            |
| 252                             | Author's response: Fixed                                                                                                                                                                                                                                                                                                    |
| 253                             |                                                                                                                                                                                                                                                                                                                             |
| 254
     | 11) "Please also note the supplement to this comment: https://www.nat-hazards-earthsyst-
sci-discuss.net/nhess-2019-67/nhess-2019-67-RC2-supplement.pdf"                                                                                                                                                                 |
| 257                             | Author's response: We responded to the comments in this pdf file, and clarify associated                                                                                                                                                                                                                                    |
| 258
            | issues in our new version of the manuscript.                                                                                                                                                                                                                                                                                |
| 260                             | Please also note the supplement to this comment:                                                                                                                                                                                                                                                                            |
| 261                             | https://www.nat-hazards-earth-syst-sci-discuss.net/nhess-2019-67/nhess-2019-67-                                                                                                                                                                                                                                             |
| 262                             | AC2-supplement.pdf                                                                                                                                                                                                                                                                                                          |
| 263                             |                                                                                                                                                                                                                                                                                                                             |
| 264                             |                                                                                                                                                                                                                                                                                                                             |
| 265                             |                                                                                                                                                                                                                                                                                                                             |
| 266                             | Referee #3                                                                                                                                                                                                                                                                                                                  |
| 267
| 1) "1- The title is irrelevant as I could not see a comprehensive seismic hazard analysis, unless the author consider defining the active and capable faults (seismic source model) is a complete hazard assessment process. It is very important component in any seismic hazard study, but it is not the entire process." |
| 272
     | Author's response: We accept this comment and have changed the title of this paper. New title: "Assessment of seismic sources and capable faults through hierarchic tectonic criteria: implications for seismic hazard in the Levant"                                                                                       |
| 275                             |                                                                                                                                                                                                                                                                                                                             |
| 276
| 2) "2- Regarding the exclusion of a very important event Md 5.8 1993 due to unreliable location, please show how much the location is uncertain. Compare this with the clear uncertainty in location around the Gulf of Aqaba."                                                                                             |

| 280
| Author's response: The event Md 5.8 1993 is indeed important, but it occurred in the most seismically active zone of the Gulf of Elat (Aqaba), in such distance from our database of mapped faults, that it would not have any impact on the current results. Since the focus in this paper is the mapping of faults, further details such as error issues of a specific earthquake, which has no impact on the mapping of any fault, is irrelevant. We now clarify this though ( lines 189-191 ).                                                |
|-----------------------------------------------|----------------------------------------------------------------------------------------------------------------------------------------------------------------------------------------------------------------------------------------------------------------------------------------------------------------------------------------------------------------------------------------------------------------------------------------------------------------------------------------------------------------------------------------------------------|
| 286                                           |                                                                                                                                                                                                                                                                                                                                                                                                                                                                                                                                                          |
| 287
                   | 3) "3- Please show how many circles are included in your calculations and the weight of each circle"                                                                                                                                                                                                                                                                                                                                                                                                                                                     |
| 290
                          | Author's response: The 'circles' were given as an illustration, as mentioned in the text. However, we now see this illustration might be confusing, so we rephrase ( lines 211-212 )                                                                                                                                                                                                                                                                                                                                                              |
| 292
                          | the continuation of the last comment: "and why did you select (calculate) such weight." Author's response: see lines 213-218 .                                                                                                                                                                                                                                                                                                                                                                                                                    |
| 294                                           |                                                                                                                                                                                                                                                                                                                                                                                                                                                                                                                                                          |
| 295
            | 4) "4- It seems that the catalog contains the aftershocks of large earthquakes, please show the role of these aftershocks as they are not due to primary tectonic movements. What is the situation if these aftershocks are removed from the calculations of earthquake kernel density distribution?"                                                                                                                                                                                                                                                    |
| 299
| Author's response: Indeed, the catalogue contains aftershocks. As already mentioned, the focus of this paper is the mapping of faults, based on the suggested methodology of hierarchic seismo-tectonic criteria. Showing the exact role of the aftershocks, and the situation if they are removed from the calculation, is a different topic. We also note that aftershocks may be also associated with reactivation of faults and even with surface ruptures, so they should not be neglected in a seismicity-based criterion for a capable fault map. |
| 306                                           |                                                                                                                                                                                                                                                                                                                                                                                                                                                                                                                                                          |
| 307
     | 5) "5- For many faults the slip rate is provided based upon geologic or GPS surveys. Such slip may contain a creep component in addition to the seismic one. The role of creep should be addressed for all active faults as it could be a source of large uncertainty."
Author's response: We had already addressed the issue of evidences for a creep component: see discussion ( lines 505-510 ) and Table 1 ( lines 845, 850 ):                                                                                                      |
| 312                                           |                                                                                                                                                                                                                                                                                                                                                                                                                                                                                                                                                          |
| 313
            | 6) "5- With the large periods of quiescence observed frequently along many parts of DST, 35 years of instrumentally recorded seismicity are very short to reflect the active tectonics accurately. This period should be extended using robust historical records."                                                                                                                                                                                                                                                                                      |

- Author's response: We regard the information of historical earthquakes for estimating the
- maximum magnitude (lines 316-318), and also consider slip rates deduces from field
- measurements of much longer periods (see Table 1, and also figure A3 we now added).
- 320

7) "6- Although the seismicity and earthquake kernel density distribution show high seismicity to
the east of the Gulf of Aqaba, neither active nor capable faults are inferred at this area."

Author's response: As we already noted (see response to Referee #2), except for few exceptions (see Sec. 3 and 6), the capable fault map does not include faults in neighbouring countries. Specifically, we mapped a few faults within the Gulf of Aqaba that we define as 'main seismic sources' (see Sec. 5). This is sufficient for our purposes, which we now describe more clearly throughout the paper.

8) "7- Abbreviations should be explained at their first appearance in the text (e.g. LRB). Some
abbreviations has no explanation (QFMI)."

- 332
- 333 Author's response: Corrected.
- 334

9) "8- Minor comments a) Line 54 contains two fullstops, please remove one of them. b) Arrange
references in line 116 in a chronological order. c) Sentence in lines 147 and 148 needs reference.
d) Change figure 4 into Fig. 4 in line 258. e) Change demonstrates into demonstrate in line 282. F)
Change is represents in line 366 into represents. g) Rewrite lines 408 to 411 as it is really so difficult
to be followed. h) Remove many in line 454. i) Sea of Galilee should be shown on a map. j) ARF is
repeated in Fig. 7. k) All the maps lack to the North Direction Indicator."

- 341
- 342 Author's response:

a) Two full-stops are now removed in all parts of the manuscript. b) Corrected (now in lines 122-123). c) We now add a new reference (now in line 155). d) Corrected (now in line 264): e) Corrected (now in line 288). f) Corrected (now in line 376). g) We rephrased (now in lines 417-420). h) Already removed due to comment by Anonymous Reviewer #1.
i) We added marks for the Dead Sea Basin (DSB) and for the Sea of Galilee (SG) to Figure 1. j) Fixed. k) All our maps are orientated such that the north is directed upwards. We do not think that a North arrow is necessary.

Assessment of potential seismic hazard for sensitive facilities by
 applying seismo-seismic sources and capable faults through
 hierarchic tectonic criteria: an example from implications for
 seismic hazard in the Levant region

Matty Sharon1,2, Amir Sagy1, Ittai Kurzon1, Shmuel Marco2, Marcelo Rosensaft1

[revised manuscript text omitted]
 that1283affected Israel and its close surroundings: J. Seismol., 20(3), 971–985, 2016.

| Fault                                                        | Strike-
Lateral slip                   | Data                          | Period                  | Reference                                                           |
|--------------------------------------------------------------|-------------------------------------------|-------------------------------|-------------------------|---------------------------------------------------------------------|
|                                                              | rate
[mm/yr]                    |                               |                         |                                                                     |
| Aragonese [ARF]                                              | <del>~5*</del>                            | <del>GPS</del>                | Recent                  | Baer et al. 1999; Hamiel
et al., 2018a                           |
| Arava [AF]                                                   | -4.9# ±0.5#
4.5±0.9!     | GPS
Geology                | Recent
37±5ka | Masson et al., 2015
Le Béon et al., 2010                         |
| Evrona [EF]
Gulf of Elat zone                             | $5.0\pm0.8\#$ $4.5\pm0.3^{*}$ (E 2.2±0.4) | GPS
GPS                    | Recent
Recent        | Hamiel et al., 2018a
Reilinger et al., 2006                      |
| Jericho -[JF]                                                | 4.8±0.7#! #
(E ~0.8)            | GPS                           | Recent                  | Hamiel et al., 2018b                                                |
| Jordan Valley
[JVF] (south)                               | 4.9±0.2#                           | Geology                       | ~48ka            | Ferry et al., 2007                                                  |
| Jordan Valley
[JVF]
( <del>centralcentre)</del> | ~ 54.9±0.3 #                       | Geology                       | ~25ka                   | Ferry et al., 2011                                                  |
| Jordan Valley
(South to Sea of
Galilee[JVF]
(north) | 4.1±0. <mark>86</mark> #&                 | GPS                           | Recent                  | Hamiel et al., 2016                                                 |
| Jordan Gorge
[JGF]                                        | 4.1±0.8#
~3#
~2.6#                  | GPS
Geology
Archaeology | Recent
~5ka
~3ka  | Hamiel et al., 2016
Marco et al., 2005
Ellenblum et al., 2015 |
| Lebanon
Restraining Bend
(LRB) zone             | 3.8±0.3*
(C 1.6±0.4)                   | GPS                           | Recent                  | Gomez et al., 2007                                                  |
| Qiryat Shemona                                               | 3.9±0.3 *!*
(E 0.9±0.4)         | GPS                           | Recent                  | Gomez et al., 2007                                                  |
| Roum [RF]                                                    | 0.86–1.05#                                | Geology                       | Holocene                | Nemer and Meghraoui, 2006                                           |
| Serghaya [SF]                                                | 1.4±0.2#                                  | Geology                       | Holocene                | Gomez et al., 2003                                                  |
| Yammuneh
(LRB northern
part)                           | <del>2.8±0.5</del>                        | GPS                           | Recent                  | Gomez et al., 2003; 2007                                            |
| Yammuneh [YF]
(north of LRB)                              | 6.9±0.1#
4.2±0.3*                      | Geology
GPS                | 2ka
Recent           | Meghraoui et al., 2003
Gomez et al., 2007                        |

**Table 1: Main strike-slip faults: average slip rate details 1285**

**Geodetic or geological measurements on a specific segment-**

10.8 mm/yr of extension normal to the fault

1287 1288 ! The upper part of the interval is preferred by the authors (field considerations)

\* According to geodetic-based model 1289

1291

\* PartiallyE, C extension and convergence, respectively, normal to the fault \* creeping from a depth of  $1.5 \pm 1.0$  km to the surface at a rate of  $2.5 \pm 0.8$  mm/yr

**1293Table 2. Marginal faults and branches with integrated slip or subsidence of ~0.5 -mm/yr1294 $\leq VS \leq \sim 1 \text{ mm/yr}$ and references**

| Fault            | Slip rate [mm/yr]     | Data       | Period       | Reference                        |
|------------------|-----------------------|------------|--------------|----------------------------------|
|                  |                       |            |              |                                  |
| Dead Sea basin   | ≥1                    | Geology    | Pleistocene- | Bartov and Sagy,                 |
| marginal faults  | Based on basin        | Geophysics | Holocene     | 2004; Torfstein et               |
|                  | subsidence rates      |            |              | al., 2009; ten Brink             |
|                  |                       |            |              | and Flores, 2012;                |
|                  |                       |            |              | Bartov and Sagy,                 |
|                  |                       |            |              | 2004                             |
| Carmel-Tirza-    | 0.9±0.45              | GPS        | Recent       | Sadeh et al., 2012               |
| Izrael fault     | -totalTotal slip rate |            |              |                                  |
| zone             | (0.7±0.45             |            |              |                                  |
| <del>[CTF]</del> | lateral; 0.6±0.45     |            |              |                                  |
|                  | extension)            |            |              |                                  |
|                  | < 0.5                 | Geology    | 200ka        | Zilberman et al.,                |
| Carmel           |                       | 0.5        |              | 2011                             |
| Hula western     | ≻~ 0.4         | Geology    | ~1 Ma        | Schattner and                    |
| border           | Based on basin        | Geophysics |              | Weinberger, 2008                 |
|                  | subsidence rates      |            |              | C .                              |
| Elat             | ?                     | Geology    | Holocene     | Amit et al., 2002;               |
|                  |                       |            |              | Porat et al., 1996;              |
|                  |                       |            |              | Amit et al., 2002;               |
|                  |                       |            |              | Shaked et al., $2\overline{004}$ |

**1296 **Figure captions**

Figure 1: -Plate configuration in the Eastern Mediterranean. Arrows show relative motion.
SR-Suez Rift; GEA: Gulf of Elat/Aqaba; DST-Dead Sea Transform fault system; CTFCarmel Tirza Fault zone; LRB-Lebanon Restraining Bend; CA- Cyprian Arc; DSB-Dead
Sea basin; SG-Sea of Galilee.

Figure 2: Epicentres in Israel and surrounding areas between the years 1983-2017, based on
 the relocated earthquake catalogue. Circle size and colours indicate the magnitude. Black
 lines represent the main fault segments of the DST and the CTF. The background for this
 figure and the followings is based on Farr et al., (2007).

Figure 3: The *earthquake kernel density* distribution, according to the relocated catalogue.
Colours and corresponding numbers indicate the value in [events/km2/yr].

Figure 4: The *seismic moment kernel density* distribution, according to the relocated 1308 catalogue. Colours and corresponding numbers indicate the value in  $log[joule/km^2/yr]$ .

Figure 5: The main seismic sources in Israel and adjacent areas. Colours indicate the two categories of faults according to the criteria. Inferred subsurface faults are marked by dashed lines. Abbreviations are for the DST main strike-slip segments, its main branches and marginal faults. Numbers indicate geodetic slip rates [mm/yr] for strike-slip components, according to recent studies (for errors and longer-term slip rates, see Tables 1,
2)-; Fig. A3). Brackets indicate slip rates accommodated by an entire fault zone. Asterisk
denotes segments of unknown slip rates, where the fault splits into a few (sub-) parallel
segments.

1317Figure 6. The seismicity polygons: earthquake density of values >  $\sim 0.001$ [events/km²/yr] and1318Mo density of values >  $\sim 9.5 log[joule/km²/yr]$ ; the product is the overlap polygon (in1319brown).

Figure 7. Quaternary fault map of Israel. Colours indicate the corresponding criterion for
each fault. Inferred subsurface faults are marked by dashed lines. Abbreviations are for the
main strike-slip segments of the DST.

1327